# Conformal Prediction Intervals with Temporal Dependence

**Zhen Lin**
*University of Illinois at Urbana-Champaign*
*Urbana, IL 61801*
`zhenlin4@illinois.edu`

**Shubhendu Trivedi**[*]
`shubhendu@csail.mit.edu`

**Jimeng Sun**
*University of Illinois at Urbana-Champaign*
*Urbana, IL 61801*
`jimeng@illinois.edu`

Reviewed on OpenReview: [https://openreview.net/forum?id=8QoxXTDcsH](https://openreview.net/forum?id=8QoxXTDcsH)

## Abstract

*Cross-sectional prediction* is common in many domains such as healthcare, including forecasting tasks using electronic health records, where different patients form a cross-section. We focus on the task of constructing *valid* prediction intervals (PIs) in time series regression *with a cross-section*. A prediction interval is considered valid if it covers the true response with (a pre-specified) high probability. We first distinguish between two notions of validity in such a setting: *cross-sectional* and *longitudinal*. Cross-sectional validity is concerned with validity across the cross-section of the time series data, while longitudinal validity accounts for the temporal dimension. Coverage guarantees along both these dimensions are ideally desirable; however, we show that distribution-free longitudinal validity is theoretically impossible. Despite this limitation, we propose *Conformal Prediction with Temporal Dependence* (CPTD), a procedure that is able to maintain strict cross-sectional validity while improving longitudinal coverage. CPTD is post-hoc and light-weight, and can easily be used in conjunction with any prediction model as long as a calibration set is available. We focus on neural networks due to their ability to model complicated data such as diagnosis codes for time series regression, and perform extensive experimental validation to verify the efficacy of our approach. We find that CPTD outperforms baselines on a variety of datasets by improving longitudinal coverage and often providing more efficient (narrower) PIs. Our code is available at [https://github.com/zlin7/CPTD](https://github.com/zlin7/CPTD).

## 1 Introduction

Suppose we are given $N$ independent and identically distributed (i.i.d) or exchangeable time series (TS), denoted $\{\mathbf{S}_i\}_{i=1}^N$. Assume that each $\mathbf{S}_i$ is sampled from an arbitrary distribution $\mathcal{P}_S$, and consists of temporally-dependent observations $\mathbf{S}_i = [Z_{i,1} \ldots, Z_{i,t}, \ldots, Z_{i,T}]$. Each $Z_{i,t}$ is a pair $(X_{i,t}, Y_{i,t})$ comprising of covariates $X_{i,t} \in \mathbb{R}^d$ and the response $Y_{i,t} \in \mathbb{R}$. Given data $\{Z_{N+1,t'}\}_{t'=1}^t$ until time $t$ for a new time series $\mathbf{S}_{N+1}$, the time series regression problem amounts to predicting the response $Y_{N+1,t+1}$ at an unknown time $t + 1$. An illustrative example is predicting the white blood cell count (WBCC) of a patient after she is

---

[*]During the initiation and pursuance of this research, the author's primary affiliation was MIT.

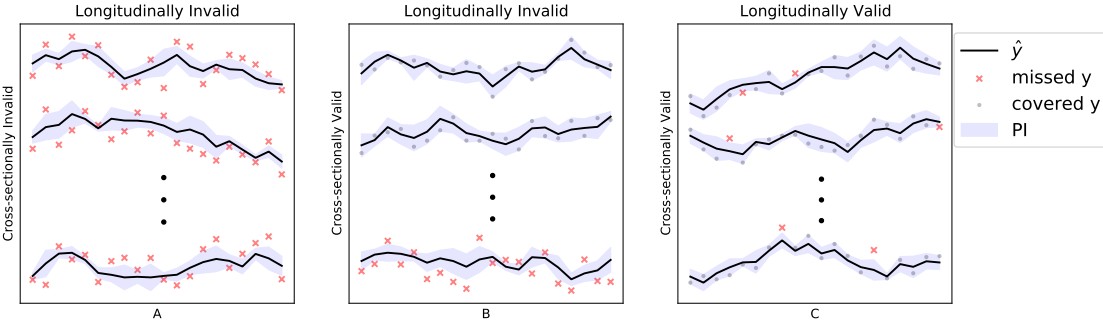

Figure 1: The figure illustrates cross-sectional validity vs. longitudinal validity, which can be seen as inter- and intra- time series coverage guarantees. The black curves are predictions by the model, and the shaded blue bands denote the PIs. Red crosses are the ground-truth $y$ not covered by PIs, while blue dots are the ground-truth $y$ which are covered. Ideally, we want a small number of red crosses (i.e., misses) that are randomly distributed across samples (cross-sectionally valid) and along the time dimension within each TS (longitudinally valid). The leftmost illustration (A) features PIs that are not valid in either sense i.e. $Y$ is never covered, neither across time series nor across time within a single time series. (B) shows a scenario with cross-sectional validity: for any $t$, the majority of TS are covered. It is however longitudinally invalid, because the PI of some TS has zero coverage. C (right) shows both cross-sectional and longitudinal validity.

administered an antibiotic. In such a case, $X_{i,t}$ could include covariates such as the weight or blood pressure of the $i$-th patient $t$ days after the antibiotic is given, and $Y_{i,t}$ the WBCC of this patient.

While obtaining accurate point forecasts is often of interest, our chief concern is in quantifying the uncertainty of each prediction by constructing valid prediction intervals (PI). More precisely, we want to obtain an interval estimate $\hat{C}_{i,t} \subseteq \mathbb{R}$, that covers $Y_{i,t}$ with a pre-selected high probability $(1 - \alpha)$. Such a $\hat{C}_{i,t}$ is generated by an interval *estimator* $\hat{C}_{\cdot,\cdot}$ utilizing available training data. We focus on scenarios with both cross-sectional and time series aspects such as electronic health record data (such as in Stankevičiūtė et al. (2021)), where different patients together form a cross-section. In such a setting there are two distinct notions of validity: cross-sectional validity and longitudinal validity. These notions are illustrated in Figure 1. Cross-sectional validity is a type of inter-time series coverage requirement, whereas longitudinal validity focuses on coverage along the temporal dimension in an individual time series. An effective uncertainty quantification method should ideally incorporate both notions satisfactorily.

In general, conformal prediction, owing to its distribution-free and model-agnostic nature, has gradually seen wider adoption for complicated models such as neural networks (Fisch et al., 2021; Angelopoulos et al., 2021; Bates et al., 2021; Lin et al., 2021; Zhang et al., 2021; Cortés-Ciriano & Bender, 2019; Angelopoulos et al., 2022). In the time series context, recent research effort, including Gibbs & Candes (2021); Zaffran et al. (2022); Xu & Xie (2021), has focused on obtaining PIs using variants of conformal prediction. However, these works invariably only consider the target TS, ignoring cross-sectional information along with the attendant notion of coverage. Moreover, such methods typically provide no longitudinal validity without strong distributional assumptions. The work of Stankevičiūtė et al. (2021), which also uses conformal prediction, is the only method that operates in the cross-sectional setting. However, Stankevičiūtė et al. (2021) ends up ignoring the temporal information while constructing PIs at different steps. On a different tack, popular (approximately) Bayesian methods such as Chen et al. (2014); Welling & Teh (2011); Neal (1992); Louizos & Welling (2017); Kingma & Welling (2014); Gal & Ghahramani (2016); Lakshminarayanan et al. (2017); Wilson & Izmailov (2020) could also be adapted to time series contexts (Fortunato et al., 2017; Caceres et al., 2021). However, such methods require changing the underlying regression model and typically provide no coverage guarantees.

A method to construct valid PIs that can handle both aforementioned notions of validity simultaneously, while preferably also being light-weight and post-hoc, is missing from the literature. In this paper, we fill this gap by resorting to the framework of conformal prediction. Our contributions are summarized as follows:

- We first dissect coverage guarantees in the cross-sectional time series setting to shed light on both cross-sectional and longitudinal validity. We show that longitudinal coverage is impossible to achieve in a distribution-free manner.
- Despite the impossibility of distribution-free longitudinal validity, we propose a general and effective procedure (dubbed Conformal Prediction with Temporal Dependence or CPTD for short) to incorporate temporal information in conformal prediction for time series, with a focus to improve longitudinal coverage.
- We theoretically establish the cross-sectional validity of the prediction intervals obtained by our procedure.
- Through extensive experimentation, we show that CPTD is able to maintain cross-sectional validity while improving longitudinal coverage.

## 2 Related Works

Work most related to ours falls along a few closely related axes. We summarize some such work below to contextualize our contributions.

**Bayesian Uncertainty Quantification** is a popular line of research in uncertainty quantification for neural networks. While the posterior computation is almost always intractable, various approximations have been proposed, including variants of Bayesian learning based on Markov Chain Monte Carlo Chen et al. (2014); Welling & Teh (2011); Neal (1992), variational inference methods Louizos & Welling (2017); Kingma & Welling (2014) and Monte-Carlo Dropout Gal & Ghahramani (2016). Another popular uncertainty quantification method sometimes considered approximately Bayesian is Deep Ensemble Lakshminarayanan et al. (2017). Bayesian methods have also been extended to RNNs Fortunato et al. (2017); Caceres et al. (2021). The credible intervals provided by approximate Bayesian methods, however, do not provide frequentist coverage guarantees. Moreover, modifications to the network structure (such as the introduction of many Dropout layers), which could be considered additional *constraints*, could hurt model performance. In contrast to such methods, CPTD comes with provable coverage guarantees. More specifically, it is cross-sectionally valid, and improves longitudinal coverage. CPTD is also post-hoc, and does not interfere with the base neural network.

**Quantile Prediction** methods directly generate a prediction interval for each data point, instead of providing a point estimate. Such methods typically predict two scalars, representing the upper and lower bound for the PIs, with a pre-specified coverage level $1 - \alpha$. The loss for point estimation (such as MSE) is thus replaced with the "pinball"/quantile loss Steinwart & Christmann (2011); Koenker & Bassett (1978), which takes $\alpha$ as a parameter. Recent works applied quantile prediction to time series forecasting settings via direct prediction by an RNN Wen et al. (2017) or by combining RNN and linear splines to predict quantiles in a nonparametric manner Gasthaus et al. (2019). Such methods still do not provide provable coverage guarantees, and can suffer from the issue of quantile crossing as in the case of Wen et al. (2017).

**Conformal Prediction (CP)**: Pioneered by Vovk et al. (2005), conformal prediction (CP) provides methods to construct prediction intervals or regions that are guaranteed to cover the true response with a probability $\geq 1 - \alpha$, under the exchangeability assumption. Recently, CP has seen wider attention and has been heavily explored in deep learning (Lin et al. (2021); Angelopoulos et al. (2021); Stankevičiūtė et al. (2021)) due to its distribution-free nature, which makes it suitable for constructing valid PIs for complicated models like deep neural networks. It is worth noting that although most CP methods apply to point-estimators, methods like conformalized quantile regression Romano et al. (2019) can also be applied to quantile estimators like Wen et al. (2017). CPTD is a conformal prediction method, but in the cross-sectional time series setting.

**Exchangeable Time Series and Cross-Sectional Validity**: The work that is most relevant to ours is Stankevičiūtė et al. (2021), which directly applies (split) conformal prediction (Vovk et al. (2005)) assuming cross-sectional exchangeability of the time series[1]. It however studies only the multi-horizon prediction setting, completely ignoring longitudinal validity. Although (cross-sectionally) valid, Stankevičiūtė et al. (2021) leads to unbalanced coverage (i.e. some TS receives poor coverage longitudinally while others high) and inefficient PIs. To the best of our knowledge, no other works explore the cross-sectional exchangeability in the context of time series forecasting.

---

[1]The authors of Stankevičiūtė et al. (2021) did not the term "cross-setional", but this is exactly what they mean.

**Long and Single Time Series and Longitudinal Validity:** Longitudinal validity is the type of validity that most works studying PI generation for time series focus on. Such works, including Gibbs & Candes (2021); Zaffran et al. (2022); Barber et al. (2022); Xu & Xie (2021), focus on the task of creating a PI at each step in a very long time series (often with over thousands of steps). For example, Gibbs & Candes (2021) propose a distribution-free conformal prediction method called ACI, which uses the realized residuals as conformal scores, and adapts the $\alpha$ at each time basing on the average coverage rate of recent PIs. To achieve distribution-free marginal validity, ACI has to (often) create non-informative infinitely-wide PIs, which is reasonable given the intrinsic difficulty stemming from the lack of exchangeability. Such methods also do not apply to our setting, because they typically require a very long window to estimate the error distribution for a particular time series as a burn-in period. Furthermore, they do not provide a way to leverage the rich information from the cross-section.

We now proceed to first discuss some preliminaries that will be required to describe CPTD in detail.

## 3 Preliminaries

Given a target coverage level $1 - \alpha$, we want to construct PIs that will cover the true response $Y$ for a specific time series, and at a specific time step, with probability at least $1 - \alpha$. However, we have not specified what kind of probability (and thus validity[2]) we are referring to. In this section, we will formally define *cross-sectional* and *longitudinal* validity, both important in our setting (See Figure 1 for an illustration). However, before doing so, we will first state the basic exchangeability assumption, a staple of the conformal prediction literature.

**Definition 1.** *(**The Exchangeability Assumption** Vovk et al. (2005))A sequence of random variables, $Z_1, Z_2, \ldots, Z_n$ are exchangeable if the joint probability density distribution does not change under any permutation applied to the subscript. That is, for any permutation $\pi \in \mathbb{S}_n$, and every measurable set $E \subseteq \mathcal{Z}^n$:*

$$\mathbb{P}\{(Z_1, Z_2, \ldots, Z_n) \in E\} = \mathbb{P}\{(Z_{\pi(1)}, Z_{\pi(2)}, \ldots, Z_{\pi(n)}) \in E\} \tag{1}$$

*where each $Z_i \in \mathcal{Z}$ (the corresponding measurable space for the random variable $Z_i$).*

Note that exchangeability is a weaker assumption than the "independent and identically distributed" (i.i.d.) assumption. We extend the definition to a sequence of random time series:

**Definition 2.** *(**The Exchangeable Time Series Assumption**) Given time series $\mathbf{S}_1, \mathbf{S}_2, \ldots, \mathbf{S}_n$ where $\mathbf{S}_i = [Z_{i,1}, \ldots, Z_{i,T}, \ldots]$, we denote $Z_{i,\{t_j\}_{j=1}^m}$ as the random variable comprised of the tuple $(Z_{i,t_1}, \ldots, Z_{i,t_m})$. Time series $\mathbf{S}_1, \mathbf{S}_2, \ldots, \mathbf{S}_n$ are exchangeable if, for any finitely many $t_1 < \cdots < t_m$, the random variables $Z_{1,\{t_j\}_{j=1}^m}, \ldots, Z_{n,\{t_j\}_{j=1}^m}$ are exchangeable.*

It should be clear that the exchangeability is "inter"-time series. Such an assumption could be reasonable in many settings of interest. For instance, collecting electronic health data time series for different patients from a hospital. Notice that Def. 2 reduces to Def. 1 when we have only one specific value of $t$. Throughout this paper, we will assume $\mathbf{S}_1, \ldots, \mathbf{S}_{N+1}$ are exchangeable time series.

### 3.1 Cross-sectional Validity

The first type of validity of PIs is what we refer to as the cross-sectional validity. This validity is widely discussed in the non-time series regression settings, often referred to as just "validity" or "coverage guaranteee" (e.g. in Barber et al. (2020)), but is rarely discussed in the context of time series regression. Cross-sectional validity refers to the type of coverage guarantee when the probability of coverage is taken over the cross-section i.e. across different points. The formal definition is as follows:

**Definition 3.** *Prediction interval estimator $\hat{C}_{\cdot,\cdot}$ is $(1-\alpha)$ cross-sectionally valid if, for any $t + 1$,*

$$\mathbb{P}_{\mathbf{S}_{N+1} \sim \mathcal{P}_S}\{Y_{N+1,t+1} \in \hat{C}_{N+1,t+1}\} \geq 1 - \alpha. \tag{2}$$

---

[2]Throughout the paper, "validity" and "coverage guarantee" are used interchangeably i.e. a "valid" PI is synonymous with a PI with "coverage guarantee".

We will sometimes use an additional subscript $_\alpha$ for $\hat{C}$ (i.e., $\hat{C}_\alpha$) to emphasize the target coverage level. As a reminder, $\hat{C}_{\cdot,\cdot}$, the estimator, denotes the model used to generate a specific PI (a subset of $\mathbb{R}$) for each $i$ and $t$.

| Symbol | Meaning |
|---|---|
| $\mathbf{S}_i = [Z_{i,1}, \ldots, Z_{i,T}]$ | Time series |
| $\mathbf{S}_{i,:t}$ | the first $t$ observations of $\mathbf{S}_i$ |
| $Z_{i,t} = (X_{i,t}, Y_{i,t})$ | Observation for the $i$-th time series at time $t$ |
| $\mathcal{P}_S$ | Distribution of $\mathbf{S}$ |
| $1 - \alpha$ | Coverage target |
| $\hat{C}_{i,t}$ | Prediction interval for $Y_{i,t}$ |
| $V(\cdot)$ or $V_{i,t}(\cdot)$ | Nonconformity score function |
| $v_{i,t}$ | Nonconformity score associated with $Y_{i,t}$ |
| $Q(\beta, \cdot)$ | $\beta$ quantile of $\cdot$ |
| $\hat{m}$ | Normalizer used in CPTD nonconformity scores |
| $g$ | A permutation invariant function (for CPTD-R) |

Table 1: Notations used in this paper

Using an example similar to one used earlier: suppose we want to predict the WBCC of a patient after the observation of some symptoms. In the first visit, there is really no time series information that can be used. Thus, the only type of coverage guarantee can only be cross-sectional. In simple terms, we could construct a cross-sectionally valid PI and say if we keep sampling new patients and construct the PI using the same procedure, about $\geq 1 - \alpha$ of the patients' initial WBCC will fall in the corresponding PI.

It might be worth a small digression here to note that the validity in Def. 3 is *marginal*. That is, the PI will cover an "average patient" with probability $\geq 1 - \alpha$. If we only consider patients from a minority group, the probability of coverage could be much lower, even if $\hat{C}$ is (cross-sectionally) valid. We direct interested readers to Barber et al. (2020) for a more thoroughgoing discussion.

## 3.2 Longitudinal Validity

Following on the above example, in later visits of a particular patient, we would ideally like to construct valid PIs that also consider information from previous visits. That is, we would like to use information already revealed to us for improved coverage, regardless of the patient. As might be apparent, this already moves beyond the purview of cross-sectional validity and leads to the notion of longitudinal validity:

**Definition 4.** *Prediction interval $\hat{C}_{\cdot,\cdot}$ is $1 - \alpha$ longitudinally valid if for almost every time series $\mathbf{S}_{N+1} \sim \mathcal{P}_S$ there exists a $T_0$ such that:*

$$t > T_0 \implies \mathbb{P}_{Y_{N+1,t}|\mathbf{S}_{N+1,:t-1}}\{Y_{N+1,t} \in \hat{C}_{N+1,t}\} \geq 1 - \alpha. \tag{3}$$

We impose a threshold $T_0$ because it should be clear that there is no temporal information that we can use for small $t$ such as $Y_{N+1,t=0}$. Here, the event $A$ being true for "almost every" $\mathbf{S}_{N+1}$ means that the probability of occurrence of $A$ is one under $\mathcal{P}_S$. Note that the crucial difference between cross-sectional validity and longitudinal validity is that the latter is similar to a "conditional validity", indicating a coverage guarantee *conditional on* a specific time series. Although highly desirable, it should be clear that this is a much stronger type of coverage. In fact, we can show that distribution-free longitudinal validity is impossible to achieve without using (many) infinitely-wide PIs that contain little information. We do so by adapting results on conditional validity, such as those in Lei & Wasserman (2014); Barber et al. (2020). We formally state our impossibility claim in the following theorem:

**Theorem 3.1.** *(**Impossibility of distribution-free finite-sample longitudinal validity**) For any $\mathcal{P}_S$ with no atom[3], suppose $\hat{C}_\alpha$ is a $1 - \alpha$ longitudinally valid estimator as defined in Def. 4. Then, for almost all $\mathbf{S}_{N+1}$ that we fix,*

$$\mathbb{E}[\lambda(\hat{C}_\alpha(X_{N+1,t+1}, \mathbf{S}_{N+1,:t}))] = \infty, \tag{4}$$

*where $\lambda(\cdot)$ denotes the Lebesgue measure. The expectation is over the randomness of the calibration set.*

At a high level, we will construct a distribution very close to $\mathcal{P}_\mathbf{S}$ except for in a small region with low probability mass. We will however require the distribution of $Y$ in this new distribution to spread out on $\mathcal{R}$. Therefore, a distribution-free $\hat{C}_\alpha$ is required to be (arbitrarily) wide as we take the limit. The actual proof is deferred to the Appendix.

---

[3]A point $s$ is an atom of $\mathcal{P}_S$ if there exists $\epsilon > 0$ such that $\mathcal{P}_S\{\{s' : d(s', s) < \delta)\}\} > \epsilon$ for any $\delta > 0$. $d(\cdot, \cdot)$ denotes the Euclidean distance.

**Remarks:** Theorem 3.1 suggests that for continuous distributions, any longitudinally valid PI estimator can only give infinitely-wide (trivial) PIs all the time. This impossibility is due to the lack of exchangeability on the time dimension. In the case of cross-sectional validity, we condition on one particular time-step, but still have the room to leverage the fact that we have exchangeable patient records to construct the PI (using conformal prediction. See Section 4). In the case of longitudinal validity, we condition on a particular patient. However, we cannot make any exchangeability assumption along the time dimension. Indeed, such an assumption would defeat the purpose of time series modeling; beside the fact that we cannot see the future before making a prediction for the past.

We should also note that Theorem 3.1 does not preclude the use of temporal information in a meaningful way. In fact, the main contribution of this paper is to incorporate temporal information to *improve longitudinal coverage while maintaining cross-sectional validity*.

## 4 Conformal Prediction with Temporal Dependence (CPTD)

### 4.1 Conformal Prediction

For the task of generating valid prediction intervals, conformal prediction (CP) is a basket of powerful tools with minimal assumptions on the underlying distribution. In this paper we will focus on the case of inductive conformal prediction (Papadopoulos et al., 2002; Lei et al., 2015) (now often referred to as "split conformal"), which is relatively light-weight, thus more suitable and widely used for tasks that require training deep neural networks (Lin et al., 2021; Kivaranovic et al., 2020; Matiz & Barner, 2019). For this section only, suppose we are only interested in PIs for $Y_{\cdot,t=0}$. We denote $Z_i = (X_{i,0}, Y_{i,0})$ and drop the $t$ subscript in $X_{\cdot,t}$ and $Y_{\cdot,t}$. In split conformal prediction, if we want to construct a PI for a particular $Y_i$, we would first split our training data $\{Z_i\}_{i=1}^N$ into a *proper training set* and a *calibration set* (Papadopoulos et al., 2002). The *proper training set* is used to fit a (nonconformity) score function $V$. We could begin with one of the simplest such scoring functions: $V(z) = |y - \hat{y}|$ where $\hat{y}$ is predicted by a function $\hat{\mu}(\cdot)$ fitted on the proper training set.

For ease of exposition and to keep notation simpler, we will assume any estimator like $\hat{\mu}$ has already been learned, and use $\{Z_i\}_i^N$ to denote the *calibration set* only. The crucial assumption for conformal prediction is that $\{Z_i\}_{i=1}^{N+1}$ are exchangeable. We could construct the PI for $Y_{N+1}$ by having

$$\hat{C}_{\alpha,N+1}(X_{N+1}) = [\hat{\mu}(X_{N+1}) - w, \hat{\mu}(X_{N+1}) + w] \tag{5}$$

$$\text{where } w = Q\left(\frac{\lceil(1-\alpha)(N+1)\rceil}{N+1}, \{\underbrace{|y_i - \hat{y}_i|}_{v_i := V(z_i)}\}_{i=1}^N \cup \{\infty\}\right), \tag{6}$$

where $Q(\beta, \cdot)$ denotes the $\beta$-quantile of $\cdot$. The $\lceil\cdot\rceil$ operation ensures validity with a finite $N$ with discrete quantiles. To simplify our discussion, we will also assume that there is no tie amongst the $\{v_i\}_i^{N+1}$ with probability 1, ensuring that there is no ambiguity for $Q$. This is a reasonable assumption for regression tasks (e.g. Lei et al. (2018)).

If the exchangeability assumption holds, then we have the following coverage guarantee (Vovk et al. (2005); Barber et al. (2022)):

$$\mathbb{P}_{Z_{N+1}}\{Y_{N+1} \notin \hat{C}_{\alpha,N+1}(X_{N+1})\} \leq \alpha \tag{7}$$

Because $Y_{N+1}$ is unknown, we typically replace $V(Z_{N+1})$ in Eq. 6 with $\infty$, which can only lead to a larger $w$ and is thus a conservative estimate that still preserves validity. The output of $V(\cdot)$ is called the nonconformity score. The absolute residual used above is one of the most popular nonconformity scores, e.g. used in Stankevičiūtė et al. (2021); Lin et al. (2021); Xu & Xie (2021); Barber et al. (2021).

### 4.2 Temporally-informed Nonconformity Scores (CPTD-M)

In this section we will describe a first attempt to improve longitudinal coverage, with a focus on the underlying intuition of the more general idea. Directly applying the split conformal method from above

(like in Stankevičiūtė et al. (2021)) ensures cross-sectional validity, but comes with an important limitation. In a sense, when a test point is queried on a calibration set, the nonconformity scores are supposed to be uniform in ranking. It is implied that the point estimates cannot be improved, for instance, when we use the absolute residual as the nonconformity score. In our task, suppose the prediction errors for a patient always rank amongst the top 5% using the calibration set up to time $t$. Even if we started assuming that this is an "average" patient, we might revise our belief and issue wider PIs going forward, or our model may suffer consistent under-coverage *for this patient*. These considerations motivate the need of *temporally-informed nonconformity scores*. We hope to improve the nonconformity score used at time $t + 1$ by incorporating temporal information thus far, making it more uniformly distributed (in ranking), so that whether $Y_{i,t}$ is covered at different $t$ is less dependent on previous cases.

We propose to compute a normalizer $\hat{m}_{N+1,t+1}$ for each $t$, and use the following nonconformity score:

$$V_{N+1,t+1}(\hat{y}, y; \mathbf{S}_{N+1,:t}) = \frac{|\hat{y} - y|}{\hat{m}_{N+1,t+1}}, \tag{8}$$

where $\mathbf{S}_{\cdot,:t}$ denotes the first $t$ observations of $\mathbf{S}_\cdot$. The idea is that if we expect the average magnitude of prediction errors for a patient to be high, we could divide it by a large $\hat{m}$ to bring the nonconformity scores of all patients back to a similar distribution. This is heavily inspired by a popular nonconformity score in the non-times-series settings —the "normalized" residual (Lei et al., 2018; Bellotti, 2020; Papadopoulos et al., 2002), where $V(z) = |\frac{y-\hat{y}}{\hat{\epsilon}}|$ and $\hat{\epsilon}$ can be any function fit on the *proper training set*. We use a simple mean absolute difference normalization strategy, or **MAD-normalization** in short, for $\hat{m}$:

$$\hat{m}_{i,t+1}^M := \frac{1}{t} \sum_{t'=1}^{t} |y_{i,t'} - \hat{y}_{i,t'}|. \tag{9}$$

The superscript $^M$ stands for MAD. One could potentially replace this simple average with an exponentially weighted moving average.

Note that one crucial difference between our $\hat{m}$ and the error prediction normalizer $\hat{\epsilon}$ lies in the source of information used. This source for $\hat{\epsilon}$ is mostly the *proper training set*, which means it faces the issue of over-fitting. This is especially problematic if there is *distributional shift*. For example, when a hospital deploys a model trained on a larger cohort of patients from a different database, but continues to its own patients as a small calibration set (which is more similar to any patient it might admit in the future). In our setting, however, conditioning on the point estimator, $\hat{m}$ does not depend on the proper training set at all. As we will see in the experiments (Section 5), $\hat{m}$ is more robust than an error predictor trained on the proper training set. We refer to this method as CPTD-M.

Once we have $\hat{m}_{N+1,t+1}$, the PI is constructed in the following way:

$$\hat{C}_{N+1,t+1}^{CPTD-M} := [\hat{y} - \hat{v} \cdot \hat{m}_{N+1,t+1}, \hat{y} + \hat{v} \cdot \hat{m}_{N+1,t+1}] \tag{10}$$

$$\hat{v} := Q\left(\frac{\lceil (1-\alpha)(N+1) \rceil}{N+1}, \left\{ \frac{|y_{i,t+1} - \hat{y}_{i,t+1}|}{\hat{m}_{i,t+1}} \right\}_{i=1}^{N} \cup \left\{ \frac{\infty}{\hat{m}_{N+1,t+1}} \right\} \right). \tag{11}$$

### 4.3 Temporally-and-cross-sectionally-informed Nonconformity Scores (CPTD-R)

In the previous section, we gave an example of incorporating temporal information into the nonconformity score. However, we still have not fully leveraged the cross-sectional data in the calibration set. In fact, even in the non-time series setting, the nonconformity score $v_i$ is not constrained to depend only on $Z_i$. All we need for the conformal PI to be valid is that the *nonconformity scores $\{V_i\}_{i=1}^{N+1}$ themselves (as random variables) are exchangeable* when $\{Z_i\}_{i=1}^{N+1}$ are exchangeable. This means that the nonconformity score can be much more complicated and take a form such as $v_i = V(Z_i; \{Z_j\}_{j=1}^{N+1})$, depending on the *un-ordered set*[4] of all $\{Z_j\}_{j=1}^{N+1}$.

---

[4]While obvious, more discussion on this can be found in Guan (2021).

It might be hard to imagine why and how one could adopt a complicated version of the nonconformity score for the non-time series case, but it is natural when we also have the longitudinal dimension. Suppose we are to construct a PI for $Y_{N+1,t+1}$ using conformal prediction, the nonconformity scores can depend on both $\mathbf{S}_{N+1,:t'}$ and the unordered data $\mathcal{S}_{:t'} := \{\mathbf{S}_{1,:t'}, \ldots, \mathbf{S}_{N+1,:t'}\}$ for any $t' \leq t+1$. To be precise, the nonconformity score $V_{N+1,t+1}$ could take the following general form:

$$V_{N+1,t+1}(\hat{y}, y) = f(\hat{y}, y; \mathbf{S}_{N+1,:t+1}, g(\mathbf{S}_{1,:t+1}, \ldots, \mathbf{S}_{N+1,:t+1})) \tag{12}$$

where $g$ satisfies the following property:

$$\forall \text{ permutation } \pi, \ g(\mathbf{S}_{\pi(1),:t+1}, \ldots, \mathbf{S}_{\pi(N+1),:t+1}) = g(\mathbf{S}_{1,:t+1}, \ldots, \mathbf{S}_{N+1,:t+1}). \tag{13}$$

Here, we propose **Ratio-to-Median-Residual-normalization** ($\hat{C}^{CPTD-R}$) as a simple example. First off, notice that while MAD-normalization can adapt to the scale of errors, it is less robust when there is heteroskedasticity along the longitudinal dimension; $\hat{m}$ will be influenced by the noisiest step $t' < t+1$. To cope with this issue, we could base $\hat{m}_{i,t+1}$ on the ranks, which are often more robust to outliers. Specifically, at $t+1$, we first compute the (cross-sectional) median absolute errors in the past:

$$\forall s \leq t, m_s := median_i\{|r_{i,s}|\} \text{ where } r_{i,s} = y_{i,s} - \hat{y}_{i,s}. \tag{14}$$

Then, for each $i \in [N+1]$, and each $t$, we compute the expanding mean of the median-normalized-residual:

$$nr_{i,t} := \frac{1}{t} \sum_{s=1}^{t} \frac{|r_{i,s}|}{m_s}. \tag{15}$$

$nr_{i,t}$ can be viewed as an estimate of the relative non-conformity of $\mathbf{S}_i$ up to time $t$. Thus, if we have a guess of the rank for $|r_{i,t+1}|$, denoted as $\hat{q}_{i,t+1}$, we could look up the corresponding quantile as:

$$\hat{m}_{i,t+1}^R := Q(\hat{q}_{i,t+1}, \{nr_{j,t}\}_{j=1}^{N+1}) \tag{16}$$

Following the notation in Eq. 12, the output of $g$, $g(\mathbf{S}_{1,:t+1}, \ldots, \mathbf{S}_{N+1,:t+1})$, is simply $Q(\cdot, \{nr_{j,t}\}_{j=1}^{N+1})$.

To obtain $\hat{q}_{i,t+1}$, we can use the following rule (expanding mean with a prior):

$$\hat{q}_{i,t+1} \leftarrow \frac{0.5\lambda + \sum_{s=1}^{t} \hat{F}_s(|r_{i,s}|)}{t + \lambda} \tag{17}$$

where $\hat{F}_s$ is the empirical CDF over $\{|r_{i,s}|\}_{i=1}^{N+1}$. For example, $\hat{F}_s(\max_i\{|r_{i,s}|\}) = 1$. Here we use $\lambda = 1$, which means our "prior" rank-percentile of 0.5 has the same weight as any actual observation. The full algorithm to compute $\hat{m}^R$ is presented in Alg.1.

With all these nuts and bolts in place, we can construct the PI $\hat{C}^{CPTD-R}$ as usual by using Eq. 10 and Eq. 11. The dependence on the $(t+1)$-th observation is simply dropped to avoid plugging in hypothetical values for $y_{N+1,t+1}$[5]. $\hat{C}^{CPTD-R}$ is somewhat complicated partially because we hope to exemplify how to let $g$ depend on the cross-section, but it also tends to produce more efficient PIs empirically (see Section 5).

We dub our general method as CPTD (Conformal Prediction with Temporal Dependence), which includes both CPTD-M and CPTD-R.

## 4.4 Theoretical Guarantees

To formally state that CPTD provides us with cross-sectional validity, we first need a basic lemma:

**Lemma 4.1.** *If $\mathbf{S}_1, \ldots, \mathbf{S}_{N+1}$ are exchangeable time series, then $\forall t$, $[V_{1,t+1}^{CPTD-M}, \ldots, V_{N+1,t+1}^{CPTD-M}]$ and $[V_{1,t+1}^{CPTD-R}, \ldots, V_{N+1,t+1}^{CPTD-R}]$ are both exchangeable sequences of random variables.*

---

[5]It could still be incorporated by performing "full" or transductive conformal prediction, which is typically much more expensive.

---

**Algorithm 1** Ratio-to-median-residual Normalization (CPTD-R)

---

**Input**:

$\{y_{i,s}\}_{i \in [N], s \in [t]}$: Response on the calibration set and the test TS up to $t$.

$\{\hat{y}_{i,s}\}_{i \in [N+1], s \in [t+1]}$: Predictions on the calibration set and the test TS up to $t+1$.

**Output**:

$\{\hat{m}_{i,t+1}\}$: Normalization factors for the nonconformity scores at $t+1$.

**Procedures**:

$\forall i \in [N+1], s \in [t]$, compute $r_{i,s} \leftarrow |y_{i,s} - \hat{y}_{i,s}|$, and $m_s \leftarrow median_i\{|r_{i,s}|\}$.

$\forall i \in [N+1]$, estimate the overall rank $\hat{q}_{i,t+1}$ using Eq. 17.

Compute the empirical distribution of the median-normalized residuals $\{nr_{i,t}\}_i^{N+1}$ using Eq. 15.

$\forall i \in [N+1]$, look-up the normalizer $\hat{m}_{i,t+1}^R$ using Eq. 16.

---

The validity for both our variants follows as a direct consequence:

**Theorem 4.2.** $\hat{C}_{N+1,t+1}^{CPTD-M}$ and $\hat{C}_{N+1,t+1}^{CPTD-R}$ are both $(1-\alpha)$ cross-sectionally valid.

All proofs are deferred to the Appendix.

**Additional Remarks**: Since the methods proposed are only cross-sectionally valid, they might raise the following natural question for some readers: What do we gain from using split-conformal, by going through the above troubles? First, we expect that the average coverage rate for the *least-covered* time series will be higher. Imagine the scenario where the absolute errors are highly temporally dependent. In such a case, CPTD-M and CPTD-R will try to capture the average scale of the errors, so that an extreme TS will not *always* fall out of the PIs (as long as such extremeness is somewhat predictable). This should be viewed as *improved* longitudinal coverage (despite the lack of guarantee). Secondly, we might observe improved *efficiency* - the PIs might be narrower on average.

Finally, one could replace $\hat{m}$ in Section 4.2 and Section 4.3, making use of a different function[6] $g$ that could potentially be more suitable for a target dataset. As long as the new $g$ satisfies Eq. 13, the cross-sectional validity will still hold. We want to emphasize that CPTD should be viewed as a general proposal to leverage both longitudinal and cross-sectional information to adjust the nonconformity score, for improved longitudinal coverage. Our focus is not on optimizing for $\hat{m}$. However, in our experiments, we found that both CPTD-M and CPTD-R already perform well despite the simple choice of $\hat{m}$.

# 5 Experiments

Through a set of experiments, we will first verify the validity of both CPTD-M and CPTD-R, as well as the efficiency (average width of the PIs). Then, more importantly, we will verify our assumption that ignoring the temporal dependence will lead to some TS being consistently under/over-covered, and that CPTD-M and CPTD-R improve the longitudinal coverage by appropriately adjusting the nonconformity scores with additional information.

**Baselines**: We use the following state-of-the-art baselines for PI construction in time series forecasting: Conformal forecasting RNN (CFRNN) Stankevičiūtė et al. (2021)), a direct application of split-conformal prediction[7]; Quantile RNN (QRNN) Wen et al. (2017); RNN with Monte-Carlo Dropout (DP-RNN) Gal & Ghahramani (2016); Conformalized Quantile Regression with QRNN (CQRNN) Romano et al. (2019); Locally adaptive split conformal prediction (LASplit) Lei et al. (2018), which uses a normalized absolute error as the nonconformity score (we follow the implementation in Romano et al. (2019)). Among the baselines, CQRNN and LASplit are existing conformal prediction methods extended to cross-sectional time series forecasting by us, and QRNN and DP-RNN are not conformal methods (and not valid).

---

[6]CPTD-M could also be viewed as having a constant $g$ that ignores the input.

[7]The authors suggest performing Bonferroni correction to jointly cover the entire horizon (all $T$ steps). This however means if $T$ ($H$ in Stankevičiūtė et al. (2021)) is greater than $\alpha(N+1)$, *all* PIs are infinitely wide Barber et al. (2022). The authors performed an incorrect split-conformal experiment, which is why the COVID19 dataset still has finite width in Stankevičiūtė et al. (2021).

**Datasets** We test our methods and baselines on a variety of datasets, including:

- `MIMIC`: Electronic health records data for White Blood Cell Count (WBCC) prediction (Johnson et al. (2016); Goldberger et al. (2000); Johnson et al. (2019)). The cross-section is across different patients.
- `Insurance`: Health insurance claim amount prediction using data from a healthcare data analytic company in North America. The cross-section is across different patients.
- `COVID19`: COVID-19 case prediction in the United Kingdom (UK) (COVID). The cross-section is along different regions in UK.
- `EEG`: Electroencephalography trajectory prediction after visual stimuli (UCI EEG). The cross-section comprises of different trials and different subjects.
- `Load`: Utility (electricity) load forecasting (Hong et al. (2016)). The original data consists of one TS of hourly data for 9 years. We split the data by the date, with different days treated as the cross-section.

`MIMIC`, `COVID19` and `EEG` are used in Stankevičiūtė et al. (2021) and we follow the setup closely. Note that for `Load`, we perform a strict temporal splitting (test data is preceded by calibration data, which is preceded by the training data), which means the *exchangeability is broken*. We also include a `Load-R` (random) version that preserves the exchangeability by ignoring the temporal order in data splitting. A summary of each dataset is in Table 2.

**Evaluation Metrics and Experiment Setup** We follow Stankevičiūtė et al. (2021) and use LSTM Hochreiter & Schmidhuber (1997) as the base time series regression model (mean estimator) for all methods. We use ADAM (Kingma & Ba (2015)) as the optimizer with learning rate of $10^{-3}$, and MSE loss. The LSTM has one layer and a hidden size of 32, and is trained with 200, 1000, 100, 500 and 1000 epochs on `MIMIC`, `COVID19`, `EEG`, `Insurance` and `Load`, respectively. For QRNN, we replace the MSE loss with quantile loss. Except for QRNN, CQRNN and DPRNN, all methods share the same base LSTM point estimator. For the residual predictor for LASplit, we follow Romano et al. (2019) and change the target from $y$ to $|y - \hat{y}|$.

We repeat each experiment 20 times, and report the mean and standard deviation of:

- Average coverage rate: $\sum_{i=1}^{M} \frac{1}{M} \overline{C}_i$ where $\overline{C}_i = \sum_{t=1}^{T} \frac{1}{T} \mathbf{1}\{Y_{i,t} \in \hat{C}_{\alpha,i,t}\}$.
- Tail coverage rate: $\sum_{j:\overline{C}_j \in L} \frac{1}{|L|} \overline{C}_j$, where $L := \{\overline{C}_i : \overline{C}_j < Q(0.1, \{\overline{C}_j\}_{j=1}^{M})\}$. In other words, we look at the average coverage rate of the *least-covered* time series. We wish it as high as possible.
- Average PI width: $\frac{1}{MT} \sum_{i=1}^{M} \sum_{t=1}^{T} \mu(\hat{C}_{\alpha,i,t})$ where $\mu(\cdot)$ is the width/length of $\cdot$.

In the above, $M$ denotes the size of the test set. All metrics here consider the last 20 steps. The target $\alpha = 0.1$ (corresponding to 90% PIs). We use the same LSTM architecture as Stankevičiūtė et al. (2021) with minor changes in the number of epochs or learning rate, except for the `Insurance` dataset where we introduce additional embedding training modules to encode hundreds of discrete diagnoses and procedures codes. In the Appendix, we include results for Linear Regression instead of LSTM.

**Results** We first report the mean coverage rates in Table 3. We see that in terms of average coverage rate, all conformal methods are valid (90% coverage) for the exchangeable datasets. In the case of `Load`, since we did not enforce exchangeability during the sample splitting, there is clear (minor) under-coverage for all conformal methods. However, CPTD is still slightly better than baselines, potentially because it can leverage information from the calibration set better. The benefits of CPTD, however, are best illustrated in Tables 4 and 5. In terms of efficiency (width), CPTD-R generally provides the most efficient valid PIs. The improvement is generally not large, but still significant. We note that for `MIMIC`, directly predicting quantiles (QRNN and CQRNN) provides more efficient PIs, which might be due to an asymmetric distribution of the prediction errors by the point-estimator. Designing temporally adjusted nonconformity scores for quantile regression (potentially based on Romano et al. (2019)) will be an interesting direction for future research. Finally, if we examine the tail coverage in Table 5, we see that both CPTD-R and CPTD-M consistently outperform baselines (with the mean width rescaled to the same). Table 5 suggests improved longitudinal coverage, which is the major focus of our paper. This can also be observed in Figure 2. The results suggest CPTD significantly improves longitudinal coverage with the temporally-adjusted nonconformity scores.

Table 2: Size of each dataset, and the length of the time series. Note that the `Insurance` dataset has up to 14 diagnoses codes, up to 17 CPT codes, and 3 other features. If we use one-hot encoding for the discrete codes, `Insurance` has 14*201+17*101+3 features instead. All results presented in this paper measures the last 20 steps, while the full results are in the Appendix.

| Properties | MIMIC | Insurance | COVID19 | EEG | Load/Load-R |
|---|---|---|---|---|---|
| # train/cal/test | 192/100/100 | 2393/500/500 | 200/100/80 | 300/100/200 | 1198/200/700 |
| $T$ (length) | 30 | 30 | 30 | 63 | 24 |
| # features | 25 | 34* | 1 | 1 | 26 |

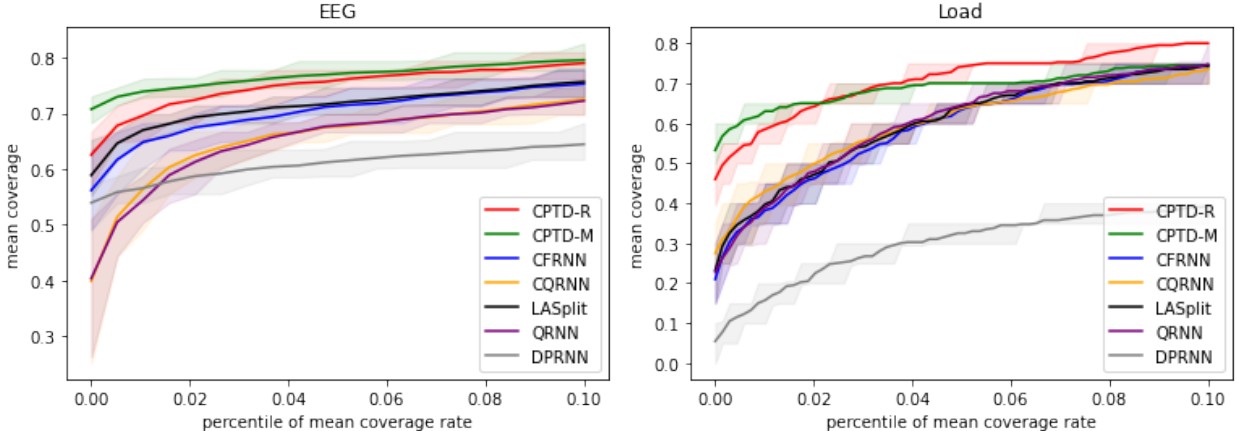

Figure 2: We show the coverage rate for the bottom 10% of the time series for `EEG` (left) and `Load` (right). All methods are re-scaled to the same mean PI width for a fair comparison. The Y-axis is the average coverage rate. The X-axis denotes the percentile among all test time series, with 0.00 meaning the least-covered time series. The band is an empirical 80% confidence band. CPTD significantly improves the longitudinal coverage rate, especially for the least-covered time series.

Table 3: Average coverage rate for each time series. Empirically valid methods are in **bold** (with p-value = 0.05). We verify that conformal prediction methods are valid, while non-conformal methods could under-cover. Note that `Load` does not satisfy the exchangeability assumption, which is why conformal methods look invalid (slightly below the target of 90%).

| Coverage ($\geq 90\%$) | CPTD-R | CPTD-M | Split (CFRNN) | CQRNN | LASplit | QRNN | DPRNN |
|---|---|---|---|---|---|---|---|
| MIMIC | **90.22±1.72** | **90.17±1.59** | **90.32±1.68** | 89.93±1.30 | **90.46±1.92** | 86.78±1.35 | 46.22±4.15 |
| Insurance | **90.01±0.63** | **90.10±0.46** | **90.05±0.64** | **90.06±0.76** | **90.05±0.72** | 85.84±0.76 | 24.56±0.76 |
| COVID19 | **90.13±1.55** | **90.27±1.07** | **90.09±1.75** | **90.08±1.61** | **90.15±1.51** | **89.18±1.52** | 68.37±3.98 |
| EEG | **89.90±1.75** | **90.08±1.48** | **89.90±1.79** | **89.96±2.26** | **89.56±1.14** | 87.94±0.94 | 38.84±1.35 |
| Load | 88.73±0.14 | 89.23±0.15 | 88.64±0.17 | 89.21±0.14 | 88.97±0.20 | 80.10±1.38 | 89.67±0.64 |
| Load-R | **90.05±0.56** | **90.17±0.73** | **90.03±0.60** | **90.23±0.62** | **90.11±0.53** | 85.35±1.06 | **90.97±0.70** |

## 6 Conclusions

This paper introduces CPTD, a simple algorithm for constructing prediction intervals for the task of time series forecasting with a cross-section. CPTD is the first algorithm that can improve longitudinal coverage while maintaining strict cross-sectional coverage guarantee. Being a conformal prediction method, the cross-sectional validity comes from the empirical distribution of nonconformity scores on the calibration set. To construct prediction intervals for $Y_{N+1,t+1}$, we propose CPTD-M, which leverages only the temporal information for the time series of interest ($\mathbf{S}_{N+1}$), and CPTD-R, which exemplifies how to use the entire calibration set to improve temporal coverage. Our experiments confirm that both CPTD-M and CPTD-R significantly outperform state-of-the-art baselines by a wide margin. Moreover, CPTD could easily be applied

Table 4: Mean of PI width. The most efficient (and valid) method is in **bold**, including methods not significantly worst than the best one. For `Load`, we show the most efficient conformal method. CPTD-R generally provides the most efficient PIs.

| Width ↓ | CPTD-R | CPTD-M | Split (CFRNN) | CQRNN | LASplit | QRNN | DPRNN |
|---|---|---|---|---|---|---|---|
| MIMIC | 1.696±0.163 | 1.876±0.209 | 1.759±0.166 | **1.560±0.140** | 1.872±0.185 | 1.407±0.130 | 0.584±0.027 |
| Insurance | **2.594±0.051** | 2.723±0.054 | 2.690±0.057 | 2.613±0.050 | 2.694±0.067 | 2.314±0.034 | 0.585±0.044 |
| COVID19 | **0.713±0.027** | 0.824±0.102 | **0.737±0.033** | 0.827±0.082 | **0.737±0.038** | 0.805±0.082 | 0.515±0.048 |
| EEG | **1.275±0.046** | **1.301±0.049** | **1.301±0.056** | 1.436±0.078 | **1.294±0.035** | 1.319±0.042 | 0.414±0.020 |
| Load | **0.200±0.004** | 0.230±0.005 | 0.209±0.004 | 0.216±0.005 | 0.213±0.005 | 0.168±0.005 | 0.569±0.008 |
| Load-R | **0.178±0.003** | 0.200±0.007 | **0.178±0.004** | 0.187±0.004 | 0.181±0.005 | 0.164±0.005 | 0.534±0.012 |

Table 5: The tail coverage rate (mean coverage rate for the least-covered 10% time series). For a fair comparison, we re-scaled all methods to have the same mean PI width (as CFRNN). Unlike average coverage rate, we want the tail coverage rate to be as high as possible. The best method is in **bold**, with the second-best underscored. Generally, both CPTD methods significantly outperform the baselines, providing better longitudinal coverage.

| Tail Coverage ↑ | CPTD-R | CPTD-M | Split (CFRNN) | CQRNN | LASplit | QRNN | DPRNN |
|---|---|---|---|---|---|---|---|
| MIMIC | 69.20±4.18 | 69.10±3.95 | 64.10±5.32 | **73.55±3.49** | 62.93±6.47 | 73.25±3.48 | 65.60±5.22 |
| Insurance | 71.13±1.92 | **72.49±1.32** | 66.03±2.15 | 68.28±2.94 | 68.22±2.29 | 64.72±2.52 | 47.82±2.92 |
| COVID19 | **70.22±5.05** | 70.47±2.47 | 63.78±6.74 | 59.75±6.30 | 67.34±4.41 | 59.81±6.61 | 52.56±6.60 |
| EEG | 67.30±4.34 | **71.09±3.73** | 64.35±4.23 | 57.02±6.01 | 66.88±2.03 | 57.07±3.41 | 51.06±2.92 |
| Load | **70.58±0.98** | 68.85±0.94 | 58.80±1.43 | 59.65±1.62 | 59.62±1.33 | 59.87±2.06 | 29.56±1.91 |
| Load-R | **73.03±1.46** | 71.36±1.36 | 68.69±1.96 | 69.61±1.33 | 69.83±2.03 | 69.42±1.85 | 31.92±2.19 |

to any model and data distribution. We hope CPTD will inspire future research in uncertainty quantification in time series forecasting with a cross-section.

## Acknowledgments

This work was supported by NSF award SCH-2014438, IIS-1838042 and NIH award R01 1R01NS107291-01. ST was partially supported by the NSF under grant No. DMS-1439786.

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

## A Proofs

### A.1 Proof for Theorem 3.1

#### A.1.1 Lemmas

We first present an established result on the impossibility of (non-degenerate) finite-sample distribution-free conditional coverage guarantee from Lei & Wasserman (2014):

**Lemma A.1.** *Let $\mathcal{P}$ be the joint distribution of two random variables $(X, Y)$. Suppose $\hat{C}_N$ is conditionally valid, as defined by the following:*

$$\mathbb{P}\{Y_{N+1} \in C_N(x)|X_{N+1} = x\} \geq 1 - \alpha \text{ for all } \mathcal{P} \text{ and almost all } x. \tag{18}$$

*Then, for any $\mathcal{P}$ and any $x_0$ that is not an atom of $\mathcal{P}$:*

$$\mathbb{P}\{\lim_{\delta \to 0} \underset{\|x_0 - x\| \leq \delta}{\text{ess sup}}\ L(C_N(x)) = \infty\} = 1. \tag{19}$$

The subscript $_N$ means $\hat{C}_N$ depends on a calibration set of size $N$, and $L(\cdot)$ is the Lebesgue measure.

A slightly stronger statement of Lemma A.1 is given by:

**Lemma A.2.** *Let $\mathcal{P}$ be the joint distribution of two random variables $(X, Y)$. Let $\mathcal{P}_U$ be the distribution of an additional random variable $U$ that $\hat{C}$ could use. Denote the joint distribution of $(X, Y, U)$ as $\mathcal{P}^+ = \mathcal{P} \times \mathcal{P}_U$, and $X^+ := (X, Y)$. Suppose $\hat{C}_N$ is conditionally valid with respect to the original $\mathcal{P}$, as defined by the following:*

$$\mathbb{P}\{Y_{N+1} \in C_N(X_{N+1}^+)|X_{N+1} = x\} \geq 1 - \alpha \text{ for all } \mathcal{P}^+ \text{ and almost all } x. \tag{20}$$

*Then, for any $\mathcal{P}^+$ and any $x_0$ that is not an atom of $\mathcal{P}$:*

$$\mathbb{P}\{\lim_{\delta \to 0} \underset{\|x_0 - x\| \leq \delta}{\text{ess sup}}\ L(C_N(x, U_{N+1})) = \infty\} = 1. \tag{21}$$

Below is a proof mostly following Lei & Wasserman (2014). First, we define $\epsilon_n$ and TV like in proof for lemma 1 in Lei & Wasserman (2014). For any pair of distributions $\mathcal{P}$ and $\mathcal{Q}$, we define the total variation distance between them as:

$$TV(\mathcal{P}, \mathcal{Q}) = \sup_A |\mathcal{P}(A) - \mathcal{Q}(A)| \tag{22}$$

For any $\epsilon > 0$, define $\epsilon_N = 2(1 - (1 - \frac{\epsilon^2}{8})^{\frac{1}{N}})$, and we will have (Lei & Wasserman (2014))

$$TV(\mathcal{P}, \mathcal{Q}) \leq \epsilon_N \implies TV(\mathcal{P}^N, \mathcal{Q}^N) \leq \epsilon. \tag{23}$$

Fix $\epsilon > 0$. Let $x_0$ be a non-atom and choose $\delta$ such that $\mathbb{P}_X\{B(x_0, \delta)\} < \epsilon_N$. Fix $B > 0$ and let $B_0 = \frac{B}{2(1-\alpha)}$. Given $\mathcal{P}^+$, define another distribution $\mathcal{Q}^+$ by

$$\mathcal{Q}^+(A) = \mathcal{P}^+(A \cap S^c) + \mathcal{U}(A \cup S) \tag{24}$$

where $S = \{(x, y, u) : x \in B(x_0, \delta)\}$, and $\mathcal{U}$ has total mass under $\mathcal{P}^+(S)$ and is uniform in $\{(x, y, u) : x \in B(x_0, \delta), |y| < B_0, u \in B(0, C)\}$. (We will see that the only thing that matters is that $Y$ is uniform in this small region). Note that $TV(\mathcal{P}^+, \mathcal{Q}^+) \leq \epsilon_N$, which means $TV(\mathcal{P}^{+N}, \mathcal{Q}^{+N}) \leq \epsilon$.

For all $x \in B(x_0, \delta)$ and all $u \in B(0, C)$, $\int_{C(x,u)} q^+(y|x)dy \geq 1 - \alpha$ implies $leb(C(x, u) \geq 2(1-\alpha)B_0 = B$. Therefore, $\mathcal{Q}^{+N}\{\text{ess sup}_{x \in B(x_0, \delta)} leb(C(x, U)) \geq B\} = 1$. Therefore,

$$\mathcal{P}^{+N}\{\underset{x \in B(x_0, \delta)}{\text{ess sup}}\ leb(C(x, U)) \geq B\} \geq \mathcal{Q}^{+N}\{\underset{x \in B(x_0, \delta)}{\text{ess sup}}\ leb(C(x, U)) \geq B\} - \epsilon = 1 - \epsilon \tag{25}$$

Lemma A.2 follows as $\epsilon$ and $B$ are arbitrary.

### A.1.2  Main Proofs

*Proof.* Now, for any $t$, we could view all previous observations (including $Y$) as the new "$X$", and $X_{i,t}$ as the "$U$". That is:

$$\mathbf{X}_i := [Z_{i,0}, Z_{i,1}, \ldots, Z_{i,t-1}] \tag{26}$$

$$U_i := X_{i,t} \tag{27}$$

$$\mathbf{X}_i^+ := [\mathbf{X}_i, U_i] \sim \mathcal{P}^+ \tag{28}$$

Then, the question is whether we could use the new $\mathbf{X}_{N+1}^+$, and $\{(\mathbf{X}_j^+, Y_{j,t}\}_{j=1}^N$ to create a prediction interval $\hat{C}$ such that

$$\mathbb{P}_{Y_{N+1,t}|\mathbf{X}_{N+1}}\{Y_{N+1,t} \in \hat{C}\} \geq 1 - \alpha. \tag{29}$$

Lemma A.2 tells us if $\hat{C}$ satisfies such coverage guarantee (note the conditioning on $\mathbf{X}_{N+1}$), we have, for all $x_0$,

$$\mathbb{P}\{\lim_{\delta \to 0} \underset{\|x_0 - x\| \leq \delta}{\operatorname{ess\,sup}} \; leb(\hat{C}(x, U_{N+1})) = \infty\} = 1 \tag{30}$$

$$\implies \mathbb{E}_{\mathbf{X}_{N+1}^+}[leb(\hat{C}(\mathbf{X}_{N+1}^+))] = \infty. \tag{31}$$

$\square$

### A.2  Proof for Lemma 4.1

*Proof.* We will show the general case that any nonconformity scores $V_{1,t}, \ldots, V_{N+1,t}$ generated Eq. 12 and Eq. 13 are exchangeable if $\mathbf{S}_1, \ldots, \mathbf{S}_{N+1}$ are, because $V^{CPTD-M}$ and $V^{CPTD-R}$ are special cases of nonconformity scores generated this way.

First of all, if we denote $V_{:,t}$ as a vector, it is clearly a row-permutation-equivariant function (denoted as $G$) on the matrix $\mathbf{S}_{:,:t}$, where the i-th row (out of $N+1$) is $[Z_{i,1}, \ldots, Z_{i,t}]$. Formally, for any permutation $\pi$ of $N+1$ elements, $V_{\pi,t} = G(\mathbf{S}_{\pi,:t})$. For each measurable subset $E$ of $\mathcal{V}^{N+1}$, define $E_\mathbf{S} \subset \mathcal{Z}^{N+1}$ as

$$E_\mathbf{S} := \{\mathbf{s}_{:,:t} : G(\mathbf{s}_{:,:t}) \in E\} \tag{32}$$

Because $\mathbf{S}_1, \ldots, \mathbf{S}_{N+1}$ are exchangeable, for any permutation $\pi$, we have

$$\mathbb{P}\{V_{:,t} \in E\} = \mathbb{P}\{\mathbf{S}_{:,:t} \in E_\mathbf{S}\} = \mathbb{P}\{\mathbf{S}_{\pi,:t} \in E_\mathbf{S}\} = \mathbb{P}\{V_{\pi,t} \in E\} \tag{33}$$

$\square$

### A.3  Proof for Theorem 4.2

*Proof.* Follows from Lemma 4.1 immediately. We provide a brief sketch here. With Lemma 4.1, we know that the following random variable $O_{N+1}$

$$o_{N+1} := \frac{|\{i \in [N] : v_{i,t} \leq v_{N+1,t}\}| + 1}{N + 1} \tag{34}$$

follows a uniform distribution on $\{\frac{i}{N+1}\}_{i=1}^{N+1}$. (Again we assume the probability of having a tie is zero, which means there is always a strict ordering.) Since

$$o_{N+1} \leq \frac{\lceil(1-\alpha)(N+1)\rceil}{N+1} \implies v_{N+1,t+1} \leq Q(\frac{\lceil(1-\alpha)(N+1)\rceil}{N+1}, \{v_{i,t+1}\}_i^{N+1}) \implies Y_{N+1,t+1} \in \hat{C}_{N+1,t+1}, \tag{35}$$

we have

$$\mathbb{P}\{Y_{N+1,t+1} \in \hat{C}_{N+1,t+1}\} \geq \mathbb{P}\{o_{N+1} \leq \frac{\lceil(1-\alpha)(N+1)\rceil}{N+1}\} \geq 1 - \alpha \tag{36}$$

$\square$

# B    Why does normalization help?

In this section, we will consider some simple scenarios and show why CPTD-M and CPTD-R can improve longitudinal coverage. While these scenarios are over-simplifications of the reality, we use them mainly to illustrate the main ideas behind CPTD more rigorously.

Suppose the error is both time-dependent and heteroskedastic in the cross-section. Formally, suppose the error $R_{i,t} := |Y_{i,t} - \hat{\mu}(X_{i,t}; \mathbf{S}_{i,:t-1})|$ is a random variable that factors out into a product $R_{i,t} = S_i E_t$ where the marginal distribution of $E_t$ is $\mathcal{P}_{E_t}$. We impose a mild assumption that $\mathbb{P}\{E_t = 0\} = 0$ for simplicity of discussion. That is, there is an intrinsic "error scale" for each time series. Note this assumption is not overly simplistic either: While the assumption in many related works (e.g. Barber et al. (2022); Xu & Xie (2021)) is that of *mild* distributional shift, we allow arbitrary distribution for $\mathcal{P}_{E_t}$.

**The case for CPTD-M**: Denote the percentile of $\frac{|y_{i,T} - \hat{y}_{i,T}|}{\hat{m}^M}$ among $\{\frac{|y_{i,T} - \hat{y}_{i,T}|}{\hat{m}^M}\}_{i=1}^{N+1}$ as $\hat{q}_{i,T}^M$. We will examine the following probability:

$$\mathbb{P}\{Y_{N+1,T} \in \hat{C}_{\alpha,N+1}^{CPTD-M} | F_S(S_{N+1}) \geq \beta\} \tag{37}$$

$$= \mathbb{P}\{\hat{q}_{N+1,T}^M \leq \frac{\lceil (1-\alpha)(N+1) \rceil}{N+1} | F_S(S_{N+1}) \geq \beta\} \tag{38}$$

for a large $\beta$ such as 0.99. This can be thought of a worst case coverage rate. For $T > 1$, we define a new random variable $E_T' = \frac{E_T}{\sum_{t=1}^{T-1} E_t}$ (which is defined with probability one). It is clear that $\frac{R_{i,T}}{\hat{m}_{i,T}^M} \sim \mathcal{P}_{E_T'}$ for all $i$. As a result, $\mathbb{P}\{\hat{q}_{N+1,T}^{CPTD-M} \leq \frac{\lceil (1-\alpha)(N+1) \rceil}{N+1} | F_S(S_{N+1}) \geq \beta\} = \frac{\lceil (1-\alpha)(N+1) \rceil}{N+1}$ for any $\beta$.

**The case for CPTD-R**: Now, suppose we also have heteroskedasticity along the longitudinal dimension. For simplicity of discussion, we assume $E_t = C_t e^{\mathcal{N}(0,1)}$ where $C_t$ is a non-random scalar for each $t$. Denote the percentile of the nonconformity scores for CPTD-R as $\hat{q}_{i,T}^R$ and that for the basic split conformal as $\hat{q}_{i,T}$. While the previous discussion still holds, consider a slightly different quantity than the above:

$$\mathbb{P}\{\hat{q}_{N+1,T}^M \leq \frac{\lceil (1-\alpha)(N+1) \rceil}{N+1} | \hat{q}_{N+1,1} < 1 - \beta\} \tag{39}$$

What would happen if, say, $C_1 \gg (\sup_{i,j} \frac{S_j}{S_i}) \sum_{t=2}^T C_t$? Essentially, if we use CPTD-M, $\hat{m}^M$ is dominated by the randomness of $E_1$. Therefore, $\hat{m}^M$ will be too small as long as $\hat{q}_{N+1,1}$ is very small, even if the target normalization constant $S_{N+1}$ is large. This is however not an issue for CPTD-R, because $C_t$ is always cancelled out.

# C    Additional Experimental Details

## C.1    Full TS

In Table 6 we show the same metrics as in the main text, but on the entire time series (instead of the last 20 steps).

## C.2    Tail coverage rate over time

We plot the average longitudinal coverage rate for the bottom 10% TS up to $t$, for each $t <= T$, in Figure 3. For clarity, we only include CFRNN and CPTD (including CPTD-M and CPTD-R). `Ideal` is the ideal scenario where the event of coverage is temporally independent for each time series. That is, each $\mathbf{1}\{Y_{N+1,t} \in \hat{C}\}$ follows a Bernoulli distribution of $p = 1 - \alpha$ (independently). We did not perform re-scaling in this plot because Ideal corresponds to an average coverage rate of $1 - \alpha$, which is why CPTD-M (which typically generates wider PIs) shows better coverage than CPTD-R. We could see that there are still gaps between CPTD and `Ideal`, which is a room of improvement for future works. It is also interesting that most of the gain in coverage seems to happen at the beginning - that is, CPTD-R and CPTD-M adapt to the "extreme" TSs in a few steps, and maintain the gain.

Table 6: Mean coverage, mean PI width, and tail coverage (re-scaled to same mean PI width) using the full time series. Valid mean coverage and the best of tail coverage and mean PI width are in **bold**. The conclusion is the same as in the main text - CPTD greatly improves the longitudinal coverage for the least-covered TSs, and maintains very efficient PIs (width).

| Coverage | CPTD-R | CPTD-M | Split (CFRNN) | CQRNN | LASplit | QRNN | DPRNN |
|---|---|---|---|---|---|---|---|
| MIMIC | **90.23±1.06** | **90.12±0.96** | **90.23±1.28** | 89.93±1.18 | **90.24±1.42** | 84.79±1.28 | 44.96±3.99 |
| Insurance | **89.97±0.45** | **90.02±0.32** | **90.03±0.56** | **90.05±0.62** | **89.99±0.48** | 86.32±0.63 | 25.15±0.69 |
| COVID19 | **89.97±1.47** | **90.22±0.94** | **90.02±1.69** | **90.11±1.36** | **90.10±1.30** | 89.16±1.35 | 65.98±2.91 |
| EEG | **89.63±0.92** | **90.11±0.77** | **89.53±1.14** | **89.83±1.21** | **89.48±0.93** | 86.64±0.70 | 35.19±1.17 |
| Load | 88.68±0.12 | 89.23±0.14 | 88.52±0.18 | 89.09±0.11 | 88.84±0.19 | 80.29±1.34 | **90.61±0.61** |
| Load-R | **90.01±0.54** | **90.12±0.69** | **89.98±0.61** | **90.27±0.60** | **90.03±0.48** | 85.69±1.10 | **91.94±0.62** |
| **Mean Width ↓** | | | | | | | |
| MIMIC | 1.767±0.127 | 2.136±0.185 | 1.808±0.138 | **1.640±0.137** | 1.913±0.147 | 1.416±0.126 | 0.575±0.022 |
| Insurance | **2.641±0.043** | 3.055±0.075 | 2.711±0.048 | **2.647±0.041** | 2.712±0.050 | 2.369±0.028 | 0.559±0.033 |
| COVID19 | **0.713±0.028** | 0.918±0.134 | **0.731±0.033** | 0.796±0.069 | **0.731±0.039** | 0.774±0.066 | 0.486±0.040 |
| EEG | **1.207±0.024** | 1.345±0.051 | **1.220±0.035** | 1.345±0.046 | **1.220±0.028** | 1.228±0.031 | 0.349±0.014 |
| Load | **0.186±0.003** | 0.224±0.004 | 0.194±0.004 | 0.200±0.004 | 0.197±0.005 | 0.156±0.005 | 0.559±0.008 |
| Load-R | **0.165±0.003** | 0.193±0.007 | **0.165±0.003** | 0.172±0.004 | **0.167±0.004** | 0.151±0.004 | 0.539±0.011 |
| **Tail Coverage Rate ↑** | | | | | | | |
| MIMIC | **73.83±2.01** | 70.15±2.89 | 68.95±2.86 | **75.78±3.37** | 68.30±3.77 | **75.02±3.93** | 69.82±4.36 |
| Insurance | **74.31±1.19** | 72.99±0.83 | 68.97±1.92 | 71.22±2.15 | 71.30±1.58 | 68.35±2.07 | 55.99±2.11 |
| COVID19 | **70.65±5.72** | **68.54±2.07** | 64.63±6.32 | 63.67±5.52 | **68.75±4.66** | 63.69±6.16 | 57.73±4.51 |
| EEG | 74.66±1.85 | **76.58±1.39** | 69.91±2.35 | 65.06±2.80 | 71.01±1.65 | 64.65±2.01 | 60.75±2.28 |
| Load | **70.74±0.74** | 69.42±0.79 | 58.53±1.27 | 59.67±1.56 | 59.68±1.31 | 60.26±1.71 | 32.98±2.08 |
| Load-R | **73.62±1.39** | 71.21±1.26 | 68.88±1.90 | 70.87±1.42 | 70.58±1.95 | 70.72±1.69 | 34.32±2.52 |

## C.3  Cross-sectional coverage over time

In Figure 4, we plot the (cross-sectional) mean coverage rate at different $t$. We can see that all conformal methods exhibit cross-sectional validity as expected, whereas the coverage rate for non-conformal methods vary greatly through time.

## C.4  Normalization Quality

In this section we compare the rank (Spearman) correlation between different quantities and the realized residuals. The quantities to consider are $\hat{m}^M$ for CPTD-M, $\hat{\epsilon}$ for LASplit, and the width of PI predicted by QRNN (CQRNN). The correlation could be considered a measure of the normalization quality. That is, if the correlation is high, the distribution of the rank of the normalized residual/nonconformity score will be closer to a uniform distribution, which will mitigate the under-coverage of "outlier patients". For each $t$, $corr_t := SpearmanCorrelation_i(q_{i,t}, |y_{i,t} - \hat{y}_{i,t}|)$ where $q_{i,t}$ should be interpreted as $\hat{m}$, $\hat{\epsilon}$ or PI width mentioned above. The (pooled) mean and standard deviation across all $t$ and seed pairs are reported in Table 7.

As we can see, both QRNN and LASplit typically have lower correlations at test time. This is simply because training errors are typically lower than test error (not over-fitting). LASplit is especially fragile to this because this effect is two-fold, for both the base point estimator and the error predictor. A similar argument can be found in Romano et al. (2019) explaining why CQR is better than LASplit in terms of efficiency. On the contrary, the correlation for CPTD-M is typically *higher* on the test set, benefiting from the same effect: With a simple expanding mean as $\hat{m}$, the point estimator's error on the test set is easier to predict than on the training set. Note also that QRNN does not actually issue a point estimate, so $\hat{y}_{i,t}$ is replaced with the middle of the PI. This means the level of correlation itself is probably not comparable, but the change from training to testing is still informative.

## C.5  Linear Regression in the place of RNN

We perform additional experiments replacing the base LSTM with a linear regression model. This model consists of $T$ sub-models, one for each $t$ (using data up to $t-1$ as input). The results are presented in Table 8.

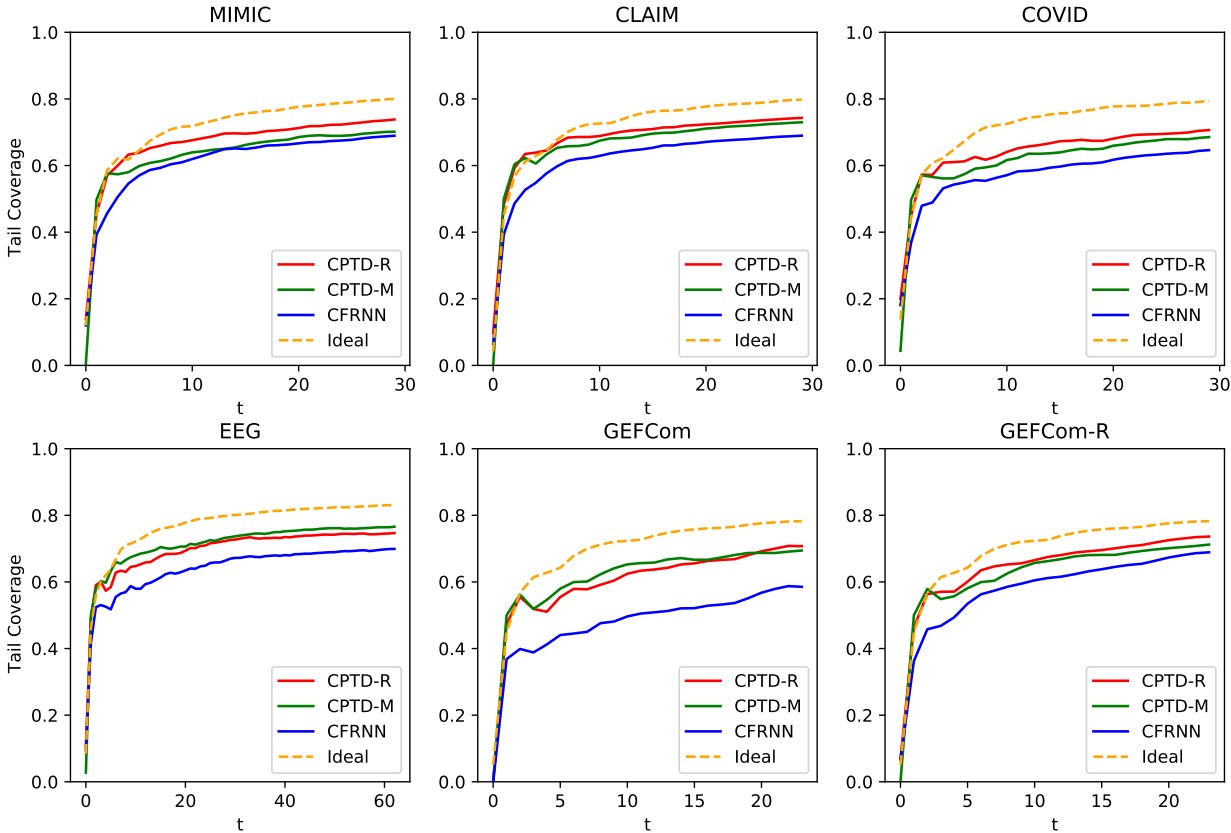

Figure 3: Tail Coverage Rate as a function of time. We plot the mean of 20 experiments. `Ideal` refers to simulated coverage events that have no temporal dependence. The X-axis is $t$, and the Y-axis is the longitudinal mean tail coverage rate up to time $t$. CPTD-M and CPTD-R typically adapt to the overall nonconformity of the TS in a few steps and maintain the advantage afterwards. There is, however, still gap between CPTD and `Ideal`.

Table 7: Normalization quality, as measured by the (cross-sectioanl) rank correlation between the realized residual and $\hat{m}^M$ for CPTD-M, PI width for QRNN, or $\hat{\epsilon}$ for LASplit.

| Rank Correlation | CPTD-M | Train QRNN | LASplit | CPTD-M | Test QRNN | LASplit |
|---|---|---|---|---|---|---|
| MIMIC | 18.87±6.29 | 35.21±7.64 | 42.28±6.85 | 24.80±9.21 | 27.69±9.86 | 14.54±10.49 |
| Insurance | 28.32±3.90 | 30.10±4.00 | 40.64±3.19 | 25.52±5.21 | 20.93±4.89 | 14.83±4.91 |
| COVID19 | 23.18±9.05 | 23.74±11.48 | 20.66±9.16 | 24.79±12.53 | 24.23±14.29 | 19.40±13.42 |
| EEG | 23.58±6.75 | 7.57±10.14 | 9.28±7.91 | 22.41±8.27 | 5.93±7.70 | 6.84±7.92 |
| Load | 9.11±5.48 | 26.63±7.50 | 24.67±9.43 | 19.29±8.78 | 26.80±10.37 | 13.38±9.02 |
| Load-R | 10.52±6.00 | 27.90±7.39 | 26.29±8.18 | 12.56±6.97 | 23.46±7.57 | 12.77±5.98 |
| Overall | 20.28±9.12 | 22.08±13.61 | 24.36±14.87 | 22.05±9.60 | 18.65±12.89 | 12.51±9.95 |

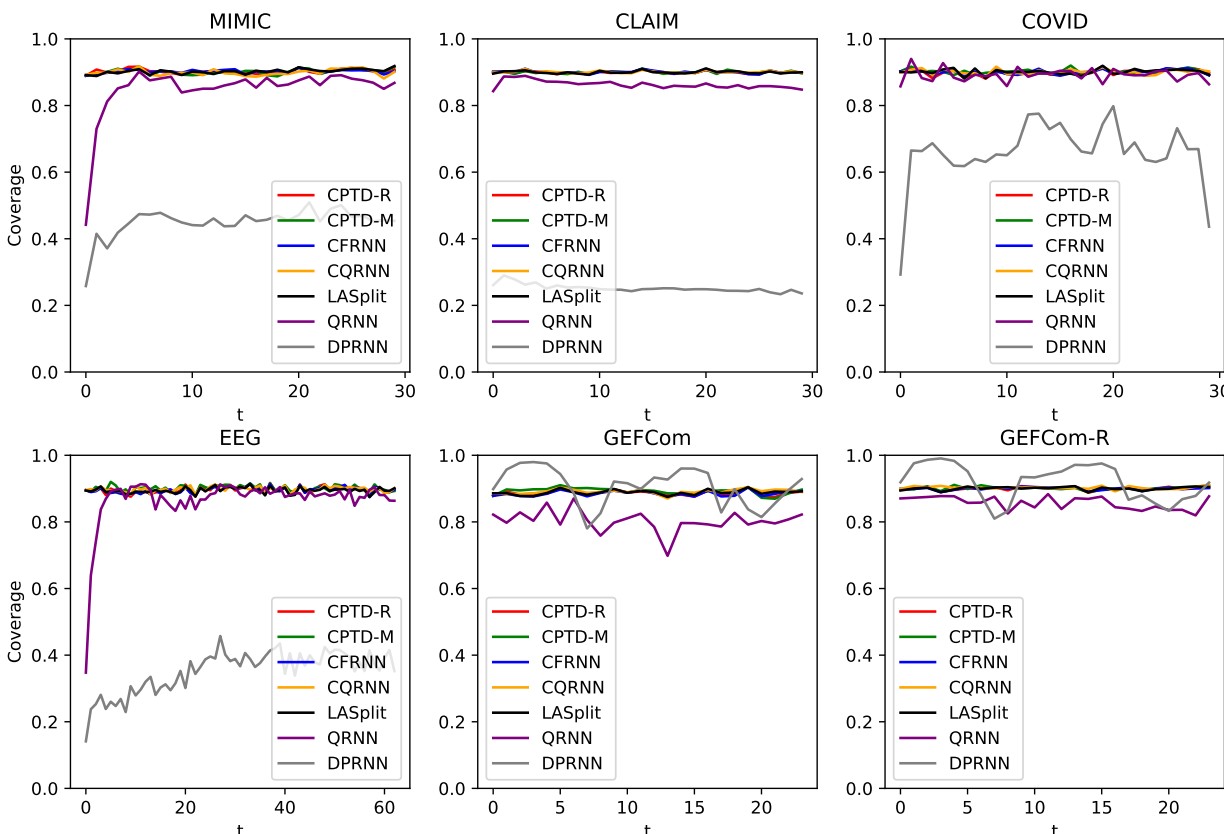

Figure 4: Mean coverage rate at different $t$. Conformal methods exhibit cross-sectional validity as expected. On the other hand, coverage rate for non-conformal methods vary greatly through time.

Table 8: Mean coverage, mean PI width, and tail coverage (re-scaled to same mean PI width) with a linear regression point estimator. Valid mean coverage and the best of tail coverage and mean PI width are in **bold**.

| Coverage | CPTD-R | CPTD-M | Split (CFRNN) | CQRNN | LASplit |
|---|---|---|---|---|---|
| MIMIC | **89.72±1.53** | **89.98±0.79** | **89.78±1.81** | **89.95±1.45** | **89.60±1.68** |
| COVID19 | **90.10±1.69** | **90.25±1.24** | **90.07±1.82** | **90.01±1.68** | **90.08±1.64** |
| EEG | **90.01±1.73** | **90.13±1.41** | **89.90±1.77** | **90.40±2.17** | **90.29±1.28** |
| Load | 89.48±0.14 | 89.86±0.12 | 89.25±0.14 | 88.93±0.26 | 89.24±0.18 |
| Load-R | **90.16±0.65** | **90.24±0.68** | **90.07±0.66** | 89.98±0.65 | **90.13±0.43** |
| Mean Width ↓ | | | | | |
| MIMIC | 2.842±0.214 | 3.115±0.334 | 2.889±0.222 | **2.608±0.227** | 2.977±0.261 |
| COVID19 | **0.780±0.019** | 0.876±0.073 | 0.808±0.031 | 0.829±0.047 | 0.807±0.026 |
| EEG | **1.632±0.082** | **1.663±0.054** | **1.653±0.091** | 1.798±0.118 | 1.810±0.070 |
| Load | **0.274±0.003** | 0.309±0.004 | **0.275±0.004** | 0.332±0.009 | 0.298±0.006 |
| Load-R | **0.266±0.009** | 0.295±0.009 | **0.267±0.009** | 0.321±0.009 | 0.284±0.007 |
| Tail Coverage Rate (Scaled) ↑ | | | | | |
| MIMIC | 64.12±4.97 | **69.05±2.24** | 60.55±5.94 | 63.80±4.38 | 62.55±5.22 |
| COVID19 | **70.16±5.80** | **71.59±2.69** | 64.16±6.86 | 66.62±4.95 | **69.06±5.52** |
| EEG | **63.46±4.09** | **66.26±2.85** | 61.44±4.65 | 52.04±7.46 | 62.79±2.53 |
| Load | **73.22±0.86** | 69.93±0.73 | 69.18±1.10 | 60.07±1.35 | 67.65±1.25 |
| Load-R | **74.36±1.26** | 71.56±1.26 | 70.95±1.09 | 63.21±1.64 | 70.16±1.23 |
| Tail Coverage Rate (Unscaled) ↑ | | | | | |
| MIMIC | 63.43±5.00 | **72.70±2.24** | 60.55±5.94 | 59.27±4.14 | 63.75±5.30 |
| COVID19 | 68.28±5.52 | **75.12±2.71** | 64.16±6.86 | 67.88±5.04 | 69.03±5.53 |
| EEG | 63.02±4.04 | **66.57±2.79** | 61.44±4.65 | 55.24±7.23 | **67.86±2.38** |
| Load | 73.15±0.85 | **75.35±0.58** | 69.18±1.10 | 69.75±1.52 | 71.39±1.09 |
| Load-R | 74.15±1.25 | **76.01±1.38** | 70.95±1.09 | 72.48±1.39 | 73.21±1.65 |

