# OpenReview forum: "Conformal Prediction Intervals with Temporal Dependence"
_TMLR — Accepted by TMLR_

### Review · Reviewer_oJ9d · 2022-06-09

**Summary Of Contributions:**


This paper proposes a conformal prediction method that can provide valid cross-sectional coverage for the time series data. The paper mathematically distinguishes cross-sectional coverage and longitudinal coverage. CPTD is proposed with a new nonconformity score design. Empirical studies show that CPTD has valid marginal coverage and improves longitudinal coverage.

**Requested Changes:**


1. Add a proof sketch of Theorem 1 to the paper. Highlight the failure mode for the longitudinal coverage. Maybe discuss the third type of coverage as mentioned in Weakness (1).

2. Have a systematic discussion on the sharpness. For example, it can be new theoretical results on the coverage upper bound and the discussion (and simulation study) on why CPTD can improve longitudinal coverage;  see Weakness (2).

3. Improve clarity:

(a) Adding implementation details in Section 4:  Are all the methods using the same neural network and training methods? If so,  $\hat{y}$ are the same for all the methods, which is the center of the predictive interval. Does this mean all methods with similar coverage would have similar interval lengths? Are the QRNN and DPRNN in Table 3 conformal methods? Add title and axis label in Figure 2.

(b) Current paper does not situate itself well in literature. The Related Works should be much earlier than just before the conclusion. Ideally, the Related Works should not just introduce existing works but describe what current methods can do, what they cannot, and what existing CPTD can solve infeasible problems. Currently, it is unclear what contributions CPTD brings to the existing literature. Some methods mentioned in "Long and Single Time-Series and Longitudinal Validity," such as Xu & Xie (2021), seem to be proper baselines.

(c) Several minor places may cause confusion: (i) Figure 1 caption "red crosses that are randomly distributed across samples". It seems the red crosses are always at the bottom sample; (ii) superscript in Eq. (9); (iii) The paragraph above Eq. (10).

**Strengths And Weaknesses:**


### Strengths:

1. One contribution of this paper is to distinguish the two types of coverages (cross-sectional and longitudinal) and show the limitations in achieving the longitudinal one. Figure 1 illustrates the difference clearly.

2. The problem of predictive inference in time series is important. The paper provides good motivation for predicting white blood cell count (WBCC) as a running example for the whole paper.

3. The simulations are conducted on a wide range of data sets and are compared with several baselines.

### Weakness:

1. The discussion of two types of coverages is an interesting contribution. That being said, I do have some questions about Theorem 1. The probability of Theorem 1 is written as $P_{Y_{N+1} | S_{N+1, 1:t-1}}$, but the complete notation should be $P_{ (X_{N+1}, Y_{N+1}) | S_{N+1, 1:t-1}}$ where $X_{N+1}$ are the covariates for the N+1 unit. In that sense, the coverage is still marginal rather than conditional at the time point level. Can the Barber et al. (2020) proof technique for the negative results on the conditional coverage be directly applied in Theorem 1? I think there is a third type of coverage here with the probability $P_{Y_{N+1} | S_{N+1, 1:t-1}, X_{N+1}}$, which corresponds to conditional coverage in the non-temporal data.

2. The empirical results of this paper are not very strong. Any valid conformal prediction method can achieve the target coverage, and an important metric for evaluation is sharpness. From Table 4, the sharpness of CPTD is on par with the existing conformal methods that can deal with time series as well. This makes the contribution less significant. One advantage might be the empirical improvement of the longitudinal coverage. But it is unclear why the algorithm design helps the longitudinal coverage, and this advantage is only shown on 2 out of 6 data sets in Figure 2.

3. The clarity of the paper needs improvement by revision. See the comments in the requested changes.

---

> ### Author Response · Authors · 2022-06-29
> **Response from authors**
>
> Thank you for the constructive suggestions, and we have made most of the requested changes.
> We however would like to clarify a few things below:
>
> ### Comment / Change 1
>
> Proof for Theorem 2.1
>
> ### Response
>
> (We assume you mean Theorem 2.1. )
> Thank you for the suggestion on including $X_{N+1,t}$ into the conditional probability. Indeed, in our previous proofs in the Appendix, we did include $X_{N+1,t}$ in the conditional. However, we updated the proof to ignore this dependence, because the impossibility result that we have here is slightly stronger without this additional conditioning. We have used the techniques in Lei and Wasserman (2014) to show this. The intuition is that as long as we are conditioning on \textit{some part} that has a probability of zero, then one can design a new ``adversarial'' distribution to force the PI estimator to get wider and wider in the small neighborhood. We have also included a proof sketch as suggested.
>
>
> ### Comment / Change 2
>
> Empirical results / Sharpness
>
> ### Response
>
> >an important metric for evaluation is sharpness
>
> While sharpness (i.e. efficiency) is indeed an important metric, the focus of this paper is longitudinal coverage. This is illustrated in Figures 2 and Table 5. We see that CPTD maintains the sharpness (Table 4) while improving longitudinal coverage significantly, which can be seen in the improved coverage for the least covered time series. CPTD achieves improved longitudinal coverage by adapting to the calibration and test time series while maintaining strict cross-sectional validity. We would also like to note that it is not very meaningful to compare the sharpness without considering the improved longitudinal coverage rate. For example, approximately conditionally valid PI estimators will likely be need to be very wide occasionally to cover outliers, whereas a marginally valid PI estimator can simply ``ignore'' them. A similar observation can be found in Section 5 of (Lei et al., 2018) for LASplit.
>
> > systematic discussion on the sharpness
>
> Thank you for the suggestion. We think that statistical accuracy has been well-studied for split conformal method (for example, (Lei et al., 2018)), and is not the focus of our work.
>
> > coverage upper bound
>
> As for the upper bound on the coverage rate - if you are referring a $\leq$ version of Equation (2), there is a classical result of $1-\alpha+\frac{1}{N+1}$ (from as early as Vovk (2005)), which is less often mentioned than the $\geq$ direction (validity).
>
>
> ### Comment / Change 2
>
> Why CPTD improves longitudinal coverage
>
> ### Response
>
> We motivated CPTD and discussed this in, for example, the first paragraph of Section 3.2. We have included a more thorough discussion on why CPTD improves longitudinal coverage in Appendix B.
>
> > this advantage is only shown on 2 out of 6 data sets in Figure 2.
>
> This is in Table 5, and CPTD outperforms baselines on most datasets. Figure 2 is just a visualization of 2 datasets.
>
> > simulation study
>
> We avoided doing a simulation study because it is easy to inject bias into such experiments. We think that real datasets (mostly taken from the CFRNN paper) provide more convincing empirical validation.
>
>
>
> ### Requested Change 3
>
> (a).  CPTD, CFRNN and LASplit share the same base LSTM (same point predictions). QRNN and DPRNN are not conformal - we have made this clearer in this update.
>
> (b). (Related Works)
> Xu \& Xie (2021) does not directly apply to our setting as it does not have a notion of cross-section (like all papers in the same section).
> For example, it bootstraps the past time steps, but our data are short time series that do not permit such bootstrapping. CPTD is the first method to consider both cross-sectional and longitudinal coverage, and we proved the results on the impossibility of (finite sample distribution free) longitudinal validity. We have tried to situate our work in the last few paragraphs of the Intro section, and the Related Works provides a more organized and complete summary of the relevant papers in the broader area.
>
> (c)(i). Red crosses in Figure 1:
> As we commented in the caption, \textbf{ideally}, we want few red crosses randomly distributed, and only Figure 1(C) is ideal.
> Red crosses are concentrated at the bottom for (A) and (B) because they are not longitudinally valid.

---

### Review · Reviewer_7oD3 · 2022-06-17

**Summary Of Contributions:**

The paper first distinguishes between cross-sectional and longitudinal validity (coverage guarantees) for time-series predictions, in a conformal prediction setting. (longitudinal means validity across the temporal access for a single time series).

It shows that meaningful longitudinal coverage is impossible to achieve in general.

Despite this, it introduces Conformal Prediction with Temporal Dependence in two variants (CPTD-M/R) and evaluates the performance compared to baselines, examining the theoretical properties (cross-sectional validity).

**Broader Impact Concerns:**

None.

**Requested Changes:**

The claims in regards to outperforming SotA baselines in the conclusion ought to be clarified slightly more: the method is better in regards to the mean of PI width and tail coverage rate.  A reader might look at table 3 and wonder whether CPTD-* are outperforming the baselines or not. Conformal methods obviously ought to have an average coverage rate according to the target $\alpha$, but this might not clear on a first glance.

## Minor requests:

WBCC in the introduction should be highlighted more or again explained later when the abbreviation is used. Took me a bit to figure out where it came from.

In eq. 6. $V(Z_{N+1})$ has already been replaced with $\infty$ (text below eq (7)) and the underbrace should only be on $|y_i - \hat y_i|$ in eq (6).

"As a proof of concept, we use..." <-- why only PoC? Does this mean CPTD-M should not be used in practice? CPTD-R which outperforms CPTD-M is also only introduced as "a simple example". Same question: if these are only PoC and simple examples, should we not use them and what should we use instead? The wording might have to be tweaked here.

In eq. (11).: why the fraction with $\infty$ and how should it be resolved?

Figure 2. Remove EEG label and add sub captions.


**Strengths And Weaknesses:**

Caveat: I am not well acquainted with the conformal prediction literature and the underlying theory, as such I cannot report on the coverage of related work or critique the theoretical results.

The paper is well-written and generally easy to understand. The figures and presentation are clean.

The breadth of empirical validation seems adequate, and the proposed methods work well.

Overall, my answers for the questions we are supposed to ask are (under above caveats):

1. Are the claims made in the submission supported by accurate, convincing and clear evidence? Yes.

2. Would at least some individuals in TMLR's audience be interested in knowing the findings of this paper? Yes.

---

> ### Author Response · Authors · 2022-06-29
> **Response from authors**
>
> Thank you very much for the constructive comments.
> We have made most of the changes, but would like to clarify some of the questions/requests.
>
> 1. ``Proof of Concept'':
> We used this wording only because these two ways to create $\hat{m}$ are the simplest/most natural that we could think of, which already work quite well. That is, we did not focus on optimizing for  $\hat{m}$, which can be done and will improve performance. We wanted to emphasize that our method should be seen as a general proposal and not tied to specific choices of  $\hat{m}$. The distinguishing characteristic of CPTD-M is that is uses the past history of a partially observed time series, and CPTD-R in addition uses the entire calibration set. However, we have removed this wording, and added a section before Section 4 clarifying this.
>
> 2. $\infty$ in Eq.(11):
> We write it this way to show where $\hat{m}_{N+1,t+1}$ is. It should not be a problem as in our case $\hat{m}$ is always finite (so the fraction equals $\infty$).

---

### Review · Reviewer_mkp5 · 2022-06-18

**Summary Of Contributions:**

This paper applies the conformal prediction approach to time-series data with the goal of providing provably valid predictive forecasts of sequential regression targets. The paper defines two notions of validity that pertain to time series data; *cross-sectional validity* which refers to the satisfaction of frequentist coverage across realizations of time series, and "longitudinal validity" which refers to coverage guarantees that hold conditioned on a realization. The authors show that a distribution-free version of the latter is impossible to satisfy.

The main contribution of the paper is a conformity score that allows for incorporation of the observed model errors in previous time steps when constructing the predictive intervals for future time steps. Unlike previous methods which ignore temporal dependencies, the two proposed methods incorporate temporal information by normalizing the conformity scores (residuals) based on aggregate information on observed residuals within the same time-series. The first variant of the proposed method (CPTD-M) applies a temporally-informed MAD normalization to each residual, and the second variant (CPTD-R) incorporates cross-sectional information as well. In both cases, the exchangeability condition is respected and hence the marginal validity guarantees of CP applies.

The experiments section demonstrates the effectiveness of the proposed method in terms of efficiency (interval width) and achieved coverage on a variety of real world data sets.

**Broader Impact Concerns:**

I do not believe there are ethical implications for this work.

**Requested Changes:**

Overall, I believe the paper is adequate for publication in TMLR conditioned on a major revision. Particularly, I encourage the authors to consider the following:

1) Providing clearer explanation on how longitudinal validity differs from conditional validity and perhaps consider modifying the terminology.

2) Adding a discussion on whether approximate longitudinal validity is achievable with the proposed methods. Ideally, a further theoretical analysis of temporally normalized residuals and whether it achieves asymptotic longitudinal validity will make this work significantly more complete.

3) Providing more discussion justifying the choice of residual normalization over other conditional approaches (such as conditional histograms, conformalized QR, localized CP, etc) for the time-series setup.

4) Expanding the set of experiments to provide other simulations (perhaps ablation experiments) that highlight the sources of performance gain from the proposed method compared to CQRNN to enable the readers to understand what information does your conformity score incorporates that CQRNN does not.

**Strengths And Weaknesses:**

The paper is well-written, rigorous and applies conformal prediction to a rather under-explored domain (time-series forecasting). I have read the paper carefully and I believe all the technical claims to be correct.

*The key weaknesses of the paper are:*

**1) [Writing-related]** I think that the notions of "cross-sectional" and "longitudinal" validity are rather unnecessary and can mislead the readers into thinking that these are conceptually different notions of validity than what is already being used in the CP literature. Essentially, cross-sectional validity is marginal validity, and longitudinal validity is conditional validity where conditioning is on filtrations of a stochastic process rather than a feature vector. Moreover, this terminology can be confusing since cross-sectional validity is required to hold for all $t$, which implies some form of longitudinal validity as well. Overall, I think validity is a probabilistic concept that should be grounded in probability measures (marginal/conditional) rather than feature dimensions (cross-sectional/longitudinal).

**2) [Theory-related]** Theorem 2.1. is identical to the results from Lei and Wasserman and its reiteration by R. Barber et al. Building on my previous comment, if we take longitudinal validity to be just a special case of the well-known conditional validity then there is no reason for this work to dedicate so much space for the same impossibility result. Moreover, I expected the paper to propose a procedure that would achieved approximate longitudinal coverage (i.e., asymptotically or up to some gap in finite samples). The fact that the following sections did not propose a method that achieve longitudinal validity in any sense makes me wonder whether the theoretical discussion of longitudinal validity was needed in the first place.

**3) [Methodological]** The proposed score is inspired by some of the earlier work on adaptive CP by normalizing residuals e.g. using a fitted location-scale model as in Lei and Wasserman, 2018. I do believe the reason this kind of construction is useful is that it enables proving approximate conditional coverage, etc, but it is not necessarily the approach that would perform the best in terms of conditional coverage and efficient. Since the authors did not show any provable benefit from the proposed score (better efficiency or provable approximate longitudinal coverage), the particular choice of the proposed score is not sufficiently justifiable. Why not just start with a model that estimates the conditional CDF given $S_{1:t}$ and then conformalize interval estimates? This would probably perform better as it conditions on the entire time-series context and not just aggregate information on residuals, which will be beneficial in heteroskedastic noise setups within a time-series (which is precisely the setup of interest for this paper). Methods that conformalize interval estimates (similar to the CQRNN that the authors use as baseline) are SOTA in non-time-series settings and they seem like a more sensible modeling choice than the proposed score.

**4) [Experiments]** While the experiments demonstrated the benefit of the proposed method, it did not fully explain why the proposed method outperforms others. Particularly, I am very surprised that the proposed score outperforms CQRNN since both methods should be incorporating the same temporal information in their scores. It is also not clear how the RNN model was conformalized and how where the hyper-parameters selected? Wouldn't some selections of hyper-parameters favor one method over the other?

---

> ### Author Response · Authors · 2022-06-29
> **Response by authors**
>
> Thank you very much for appreciating our work and your constructive comments. Due to space constraints we will split our responses into two comments.
>
> ### Writing: Cross-sectional/longitudinal validity
>
> We use these definitions to convey the difference in two types of coverage that matter in reality, and they expose a limitation of the first work in this area (CFRNN). We observed that some patients are under-covered consistently, and we tried to summarize this with the notion of longitudinal validity. This is not purely about "feature dimensions". The key reason that we used these two definitions is that there is a fundamental asymmetry between the time dimension and the cross-section. For example, we can say "probability taken over" $\mathcal{S}_{N+1} \sim \mathcal{P}_S$, but $t \sim \mathcal{P}_T$ won't make sense.
>
> ### Theory
>
> First, as noted by reviewer oJ9d, the result from Lei and Wasserman *cannot* be directly used to show our Theorem 2.1. We have since updated our proof by modifying their proof, in the Appendix.
>
> > there is no reason ... to dedicate so much space for the same impossibility result
>
> Let us explain the motivation for the result: We are the first to notice the issue of low longitudinal coverage in the setting of cross-sectional time series forecasting common in domains like healthcare. Our contribution is mainly dissecting the two notions and applying (a updated version) of the conditional validity result to TS forecasting. We think that the result also serves as a good warmup after we have introduced the two notions. We would also like to note that the similarity with conditional validity is on the surface, and treating it as such will lead to many issues (e.g. we would need one model/kernel for each $t$ in the "conformal prediction with localization" setting (Lin et al.2021 or Guan 2021)). We spent about half a page to explain why longitudinal validity is impossible to draw a limit for CPTD or any similar efforts.
>
> > approximate longitudinal coverage (i.e., asymptotically or up to some gap in finite samples)
>
> Asymptotic or approximate longitudinal validity can be seen discussed in the part ``Long and Single Time-Series and Longitudinal Validity'' (e.g. in (Barber et al. 2022) and ACI (Gibbs & Candes 2021)). In general, such validity is impossible without strong assumptions, and we do feel that this is beyond the scope of our paper. Moreover, like CFRNN, we do not have very long time series which renders asymptotic longitudinal validity not really useful. The only case of asymptotic (marginal) longitudinal coverage guarantee that we know is ACI. ACI achieves it via infinitely-wide intervals, and one could easily come up with a distribution such that ACI PIs cover the response \textit{only if} the PI is infinitely wide. (Other theoretical guarantees in ACI assume a stationary HMM).
>
> > up to some gap in finite samples
>
> Similarly, (Barber et al. 2022) suggests that the $d_{TV}$ and thus the bound for coverage gap becomes one (i.e. becomes degenerate), if the weights are data dependent and the data is time-dependent (which is the true for all our data). With these considerations in mind, the focus of our paper was always maintaining *strict* cross-sectional validity while *improving* longitudinal coverage.
>
> ### Method
>
> > did not show any provable benefit from the proposed score
>
> With distribution assumptions (see discussion above), we added Appendix B for this discussion.
>
> > Why not ... estimates the conditional CDF ...
>
> Estimating the entire CDF is nontrivial as far as we know, especially if we want to avoid quantile crossing, and it might require more training data than estimating a specific quantile (QRNN). This is surely an interesting method to our problem, but this could be addressed in an independent work proposing a new method for time-series forecasting in our opinion.
>
> > Methods that conformalize interval estimates ... are SOTA in non-time-series settings
>
> Methods like CQR perform well in the non-time-series setting *in terms of efficiency*, but it is not clear why this should/will translate to improved longitudinal coverage. Such methods also have inherent challenges, such as quantile crossing, or that the point estimates are outside of the PIs.
> This is probably why residual-based methods are still quite popular (e.g. CFRNN). Thus, even if an interval based method could tuned to outperform CPTD on some datasets, CPTD still has its own value.

---

> > ### Author Response · Authors · 2022-06-29
> > **Responses continued**
> >
> > We continue our comments here:
> >
> > ### Experiment
> >
> > >  both methods should be incorporating the same temporal information in their scores
> >
> > Actually they do not.  As we discussed before Eq.(10), baselines *cannot* leverage information from the calibration set like CPTD-R.
> > This is the main motivation of Section 3.3.
> >
> > > it did not fully explain why the proposed method outperforms others
> >
> > We would like to emphasize again that we only know CQR is good in *efficiency* in non-time-series settings. We think this prior observation does not contradict our experiment results. As noted above, CPTD-R leverages more information from the calibration set, and we saw that CPTD-R>CQR>CPTD-M in general in terms of efficiency.  It is also possible that after more careful consideration of some design choices and tuning, CQR+QRNN will outperform CPTD-R on some datasets (already so on MIMIC) in terms of efficiency.
> >
> > However, it is not clear why we should be able to go with the intuition that CQR will also be good for longitudinal coverage, as this is not a concept present in non-time-series settings. We would also like to point out that, in fact, we are the first to apply CQR to time series forecasting to our best knowledge. In other words, there is no such a baseline paper proposing CQRNN (only CFRNN).
> >
> > To address your comment, we computed the cross-sectional rank correlation to the realized residual $|y_{\cdot, t+1} - \hat{y}_{\cdot, t+1}|$ for the normalization factor $\hat{m}^{M}$ in CPTD-M. We also computed such correlation for the width of QRNN, and the error prediction by LASplit.  Basing on our motivation (to create a more uniformly distributed nonconformity score), the longitudinal coverage should be improved if this correlation is higher. We found that the training correlation for CPTD-M/(C)QRNN/LASplit are 20.28/22.08/24.36, and the test correlations are 22.05/18.65/12.51 - only CPTD-M sees an improvement. This makes sense as training errors are usually lower than test error (a similar argument can be found in the CQR paper explaining why CQR is better than LASplit). This hurts LASplit more than QRNN, because both the point estimator and the error predictor could see this effect. On the other hand, it helps CPTD-M, because the test errors are easier to predict than training errors.
> >
> > > how the RNN model was conformalized
> >
> > CFRNN is just split conformal.
> >
> > > how where the hyper-parameters selected
> >
> > We do not have many to tune/select. We tried to make everything simplistic so as to convey main proposal (to use cross-section and temporal information). Thus, for CPTD-M, the normalization is a simple average instead of EWMA (which will permit some additional tuning). The optimizer is the default Adam optimizer form PyTorch. The LSTM architecture is taken from CFRNN. The only thing we needed to set is the number of training epochs. We set the training epochs to the first ``round'' number (e.g. 100, 200) after the base LSTM converges. The losses are very flat after. We did not find much sensitivity to these parameters -- the loss becomes flat soon after and the results look similar, so we ended up using these.
> >
> >
> > ## Requested Changes 1:  how longitudinal validity differs from conditional validity
> >
> > The difference has been pointed out by reviewer oJ9d, and we discussed in mostly in  "Writing" and "Theory above. We have also updated our proof accordingly.
> >
> > ## Requested Changes 2: Approximate/asymptotic longitudinal coverage
> >
> > As discussed above (in "Theory"), this is not possible without strong distributional assumptions. We however included Appendix B with such distributional assumption. We would also like to point out that our setting is fundamentally different from "Long and Single Time-Series and Longitudinal Validity" (see section 5) so asymptotic longitudinal coverage is less meaningful in practice.
> >
> > ## Requested Changes 3: Justify the normalization choice
> >
> > We have added Appendix B for this purpose. We would like to emphasize that CPTD is proposing a *framework* to leverage both past predictions and the cross-section (calibration set), *given the trained model*. The setting is very common in healthcare, for example, because hospitals won't share patient data, and might only have access to a trained (commercial) model and its own, typically smaller, patient database. As suggested in the paper, under the framework of CPTD, one could replace MAD with EWMA of residuals or something fancier. We have thus updated the ``Remarks'' in Section 3.4 to clarify this.
> >
> > It is also worth noting that *even if* the performance is similar, most alternatives suggested (such as conformalized QR) are less flexible (i.e. if we are a hospital given a trained RNN with no access to the original data).
> >
> > ## Requested Changes 4: More experiments
> >
> > We have included more results in the Appendix (C.4) to explore why CPTD outperforms CQR in our datasets.

---

> > > ### Author Response · Authors · 2022-06-29
> > > **Responses continued**
> > >
> > > > ablation experiments
> > >
> > > We feel the current experiments are already equivalent to an ablation study in the following sense:
> > > CPTD only introduces a $\hat{m}$.
> > > If removed, it becomes CFRNN.
> > > If replaced with an error prediction, it becomes LASplit.
> > > For CQRNN, if we remove the conformalization step, it becomes QRNN.
> > >
> > > > what information does your conformity score incorporates that CQRNN does not
> > >
> > > See discussion before Eq. (10).

---

### Decision · Action_Editors · 2022-08-05

**Recommendation:** Accept as is

**Comment:**

The paper investigates conformal prediction for time series data. The authors define two notions of validity, namely cross-sectional validity and longitudinal validity. The authors propose "conformal prediction with temporal dependence" (CPTD), a method to maintain cross-sectional validity while improving longitudinal coverage. The reviewers found the paper well-written and interesting. Experimental results demonstrate the effectiveness of CPTD on a variety of real world datasets. The authors satisfactorily addressed the reviewer comments and all the reviewers lean towards acceptance.

I recommend accept. Congratulations on the nice work!